



# Data-driven versus self-similar parameterizations for Stochastic Advection by Lie Transport and Location Uncertainty

Valentin Resseguier[1], Wei Pan[2], and Baylor Fox-Kemper[3]

[1]Lab, SCALIAN DS, Espace Nobel, 2 Allée de Becquerel, 35700 Rennes, FRANCE
[2]Department of Mathematics, Imperial College London, London SW7 2AZ, UK
[3]DEEPS and IBES, Brown University, Providence, RI 02912, USA

**Correspondence:** Valentin Resseguier (valentin.resseguier@scalian.com)

**Abstract.** Stochastic subgrid parameterizations enable ensemble forecasts of fluid dynamics systems and ultimately accurate data assimilation. Stochastic Advection by Lie Transport (SALT) and models under Location Uncertainty (LU) are recent and similar physically-based stochastic schemes. SALT dynamics conserve helicity whereas LU models conserve kinetic energy. After highlighting general similarities between LU and SALT frameworks, this paper focuses on their common challenge: the

parameterization choice. We compare uncertainty quantification skills of a stationary heterogeneous data-driven parameterization and a non-stationary homogeneous self-similar parameterization. For stationary, homogeneous Surface Quasi-Geostrophic (SQG) turbulence, both parameterizations lead to high quality ensemble forecasts. This paper also discusses a heterogeneous adaptation of the homogeneous parameterization targeted at better simulation of strong straight buoyancy fronts.

## 1   Introduction

Geophysical fluid dynamics and other turbulent flows cover a wide range of spatial and temporal scales (see, e.g., Fox-Kemper, 2018). However, numerical simulations and observations are limited in the range of scales that can be simulated or observed. Accordingly, deterministic subgrid models or parameterizations (e.g. Margolin et al., 2006; San et al., 2013; Bachman et al., 2017; Voosen, 2018) and regularizations (e.g. nugget in optimal interpolation, Tandeo et al., 2014) are incorporated into numeri-

cal simulations and measurement processes. The limited resolution of these numerical simulations introduces unknown errors which are continuously amplified during the course of the simulation and which compound with the measurement errors, to the extent that the simulations model the true chaotic nature of fluid dynamics. Improvements in accuracy of these simulations can sometimes be made by judicious use of data assimilation (DA) techniques. In the data assimilation process, observations (data) are integrated into the model state of the numerical simulation limited by imperfect physics and imperfect initial conditions.

DA proceeds by alternating between forecast and analysis cycles. In each analysis cycle, observations are combined with the results from a prediction model (the forecast) to produce a so-called analysis. The analysis is expected to be more accurate than either predictions based on physical models without incorporating observations, or predictions from physics-free interpolations





of the observations. In order to optimize the analysis cycle, confidence in both the observations and the prediction needs to be quantified. Observational accuracy depends on the instrumental precision and sampling; and well-known uncertainty quantifi-

cation (UQ) techniques are available to estimate the uncertainty associated with these limitations. In contrast, the uncertainty due to imperfect model physics is more difficult to quantify because often a "perfect model", exact solution, or direct numerical simulation is not available or feasible for the intended application.

Historically introduced to address the limited range of scales simulated (e.g. Orszag, 1970; Leith, 1971), stochastic subgrid parameterization is now a widely-used UQ tool for assessing the impact of imperfect physics in numerical simulations. In

particular, it is generally thought that building such stochastic parameterizations from physical principles promises accuracy and robustness (Berner et al., 2015). Several authors have addressed the issue of stochastic subgrid parameterization evaluation through averaging or homogenization procedures (e.g. MTV method, see Majda et al., 1999). Correlated additive and multiplicative noises usually play a key role in these approaches (e.g. Gottwald and Melbourne, 2013). Sardeshmukh and Sura (2009); Sardeshmukh et al. (2015); Pearson and Fox-Kemper (2018) highlight the relevance of skew-symmetric and multiplica-

tive noises in geophysical fluid dynamics applications. Mimetic symmetries built into the fluid dynamics and stochastic subgrid model formulations may preserve some important physical invariants – a desired properties of many UQ models (e.g. Frank and Gottwald, 2013; Sapsis and Majda, 2013; Mitchell and Gottwald, 2012; Majda, 2015) and DA algorithms (Janjić et al., 2014). Other stochastic subgrid parameterizations involve memory terms, justified by the Mori-Zwanzig equations (Chekroun et al., 2011; Lu et al., 2017).

The dynamics under location uncertainty (LU) is a type of systematic stochastic subgrid parameterization introduced by Mikulevicius and Rozovskii (2004b) and Flandoli (2011) for the theory of well-posedness of stochastic partial differential equations, and by Mémin (2014) for more applied purposes. More theoretical results about models under location uncertainty are discussed in Resseguier et al. (2017a). This framework has been successfully applied to Uncertainty Quantification (Resseguier et al., 2017b), geophysics (Resseguier et al., 2017c), attractor exploration (Chapron et al., 2018) and optical flow

(Cai et al., 2018).

A similar class of models called Stochastic Advection by Lie Transport (SALT) was first derived in Holm (2015) for the time homogeneous subgrid parameterization and extended to the inhomogeneous case in (Gay-Balmaz and Holm, 2018). In the following, we will refer to this class of models as SALT. The approach used is Hamilton's variation principle with the constraint that fluid parcels move according to a Eulerian stochastic subgrid model, see (Holm, 2015; Cotter et al., 2017). A

local well-posedness result for the 3D SALT Euler equations is proved in Crisan et al. (2017) and a global well-posedness result for the 2D case is proved in Crisan and Lang (2018). Numerical studies for the implementation of the SALT subgrid parameterization are discussed in (Cotter et al., 2018b, a) for a 2D Euler model and a two-layer 2D QG dynamics. A first study on employing SALT dynamics for data assimilation is provided in Cotter et al. (2019).

In both LU and SALT frameworks, the velocity is decomposed into a random large-scale component, $\boldsymbol{w}$, and a time-

uncorrelated component, $\boldsymbol{\sigma}\dot{\boldsymbol{B}} = \boldsymbol{\sigma}\mathrm{d}\boldsymbol{B}_t/\mathrm{d}t$. The latter is Gaussian (conditionally on some large-scale quantities), correlated in space, with possible heterogeneities and anisotropy. Hereafter, this unresolved velocity component will further be assumed to be solenoidal. To parameterize those spatial correlations, we apply an infinite-dimensional linear operator, $\boldsymbol{\sigma}$, to





a $d$-dimensional space-time white noise, $\dot{\boldsymbol{B}}$. Holm and coauthors rely on a different but equivalent notation. It directly explicits the spectral decomposition of the time-uncorrelated velocity spatial covariance. Specifically, the small-scale velocity reads

$\boldsymbol{\sigma}(\boldsymbol{x})\dot{\boldsymbol{B}} = \sum_p \boldsymbol{\xi}_p(\boldsymbol{x})\mathrm{d}W_p(t)/\mathrm{d}t$ where $(W_p)_p$ is a set of independent one-dimensional Brownian motions and $(\boldsymbol{\xi}_p)_p$ a set of orthogonal functions.

That unresolved velocity is the central object of both SALT and LU approaches. The statistics of this velocity – defined either by $\boldsymbol{\sigma}$ or $(\boldsymbol{\xi}_p)_p$ – constitutes the parameterization of the random model. In this paper, we discuss several choices for this parameterization, and dedicate significant discussion to the physical principles and assumptions about the flow regime

underlying each choice. To better motivate this work, we will highlight the strong links and some formal difference between the LU and the SALT approaches. However, a full – theoretical and numerical – comparison of LU and SALT is beyond the scope of this paper.

The paper is structured as follows.

- Section 2.1 focuses on parametric non-data-driven choices. Specifically, we propose a non-stationary and tuning-free
improvement of the work of Resseguier et al. (2017b) based on self-similar assumptions and possible heterogeneous modulation. Links with the Smagorinsky subgrid parameterization and non-linear energy fluxes are also discussed.

- Section 2.2 discusses a data-driven stationary but possibly heterogeneous parameterization implemented by Cotter et al. (2018b, a). The method relies on empirical orthogonal functions (EOF) decomposition also known as proper orthogonal decomposition (POD) or principal component analysis (PCA).

- Section 3 begins by presenting the deterministic and stochastic Surface Quasi-Geostrophic models, their numerical setup and simulation parameters. These are followed by detailed discussions of uncertainty quantification (UQ) skills in Sect. 3.4.

- Sections 4 and 5 concludes this work.

Additionally, in the appendix we include a mathematical summary of LU and SALT formulations for the interested readers.
The aim here is to highlight the similarities and differences between the two approaches. Their differing sets of invariants are listed.

The choice of Surface Quasi-Geostrophic (SQG) flow is convenient since the SALT and LU versions of the SQG dynamics coincide, thus the results presented apply to both SALT and LU models. Furthermore, present interest in oceanic submesoscale motions for which SQG dynamics is often a reasonable approximation (e.g., Lapeyre and Klein, 2006b; Callies et al., 2016),
and the fact that SQG offers the possibility of reconstructing subsurface flow properties from surface satellite observations (e.g., LaCasce and Mahadevan, 2006; Isern-Fontanet et al., 2006, 2008; Klein et al., 2009; Wang et al., 2013), have results in a recent surge in the use of the SQG equations for geophysics. Accordingly, the discussion can focus on the parameterization choices in this timely application.





## 2 Parameterization of SALT-LU models: the statistics of the unresolved velocity

### 2.1 Parametric and self-similar model for the unresolved velocity


In this section, we propose parametric non-data-driven models for the unresolved velocity statistics. Similar to the Kraichnan's model (Kraichnan, 1968, 1994; Gawedzky and Kupiainen, 1995; Majda and Kramer, 1999), Resseguier et al. (2017b) introduce an unresolved velocity which is homogeneous and isotropic. Even though this method leads to good numerical results, some parameters need to be tuned. To overcome this drawback, we propose here a parameter-free improvement based on what we call

the Absolute Diffusivity Spectral Density (ADSD). New subgrid statistics will be defined at each time step from the resolved velocity kinetic energy spectrum. Finally, we also propose a heterogeneous modulation of the previous method based on the local energy flux and discuss the link with Smagorinsky-like subgrid parameterizations.

#### 2.1.1 Parameter-free and non-stationary homogeneous model

We first define the statistics of the small-scale velocity – through what we call the ADSD. Then, we explicit how to sample the

unresolved velocity from this ADSD.

In Resseguier et al. (2017b), the absolute diffusivity (i.e. KE times correlation time (Falkovich et al., 2001; Penland, 2003; Klyatskin, 2005; Vallis, 2006; Keating et al., 2011)) of the unresolved velocity is twice the variance tensor trace $\mathrm{tr}(\boldsymbol{a}) = \mathbb{E}\|\boldsymbol{\sigma}\mathrm{d}\boldsymbol{B}_t\|^2/\mathrm{d}t$ whereas the unresolved kinetic energy is $\mathrm{tr}(\boldsymbol{a})/\mathrm{d}t$. Clearly, this kinetic energy has no physical meaning. Indeed, it depends on the simulation time step and one should have the possibility to choose the time step as close as possible

to zero. Thus, the unresolved velocity amplitude is specified through an absolute diffusivity rather than through a KE. In the mathematics literature of homogenization, Kubo-type formulas may be seen as what physicists call absolute diffusivities. More generally, since the variance of a time-continuous white noise is infinite, it is more relevant to deal with absolute diffusivity rather than kinetic energy in order to describe the statistics of the time-uncorrelated velocity. Thus, keeping a spectral approach, we define – for any spatio-temporal field – an Absolute Diffusivity Spectral Density (ADSD) denoted $A(\kappa)$ at the wave-number

$\kappa$. We will rely on this ADSD rather than on the KE spectrum, $E(\kappa)$. Since the absolute diffusivity is the variance multiplied by the correlation time, it is naturally to defined the ADSD as follows:

$$A(\kappa) \triangleq E(\kappa)\tau(\kappa) \text{ where } \tau(k) = \frac{1/\kappa}{v_\kappa} = \frac{1/\kappa}{\sqrt{\kappa E(\kappa)}}, \tag{1}$$

is the eddy turnover time at the scale $1/\kappa$ and $v_\kappa$ is at characteristic velocity a this scale. Accordingly, we have:

$$A(\kappa) = \kappa^{-3/2}E^{1/2}(\kappa). \tag{2}$$

If in addition we assume a KE self-similar distribution,

$$E(\kappa) = C^2\kappa^{-s}, \tag{3}$$

we obtain:

$$A(\kappa) = C\kappa^{-r}, \tag{4}$$





where $r = (s + 3)/2$.


We aim at defining the unresolved velocity ADSD from the large-scale velocity. For this purpose, we will assume the self similar model (4) is valid at all spatial scales. At each time step, we compute the ADSD of the large-scale velocity, $A_{\boldsymbol{w}}$, from formula (2). Then, we fit the coefficients $C$ and $r$ of Eq. (4), as illustrated by figure 1. Let us denote with $C_{\boldsymbol{w}}$ and $r_{\boldsymbol{w}}$ these coefficients. Note that they are time-dependent because they depend on $\boldsymbol{w}$ which is. More precisely, we estimate the coefficients

$C_{\boldsymbol{w}}$ and $r_{\boldsymbol{w}}$ in a wavenumber interval which approximately represents a inertial range of fully-resolved scales (i.e. before the spectrum roll-off). In the left panel of figure 1, this interval is delimited by two vertical lines.

From there, we can define the unresolved velocity ADSD in such a way that the total velocity – resolved plus unresolved – meets (4) at small spatial scales with the same coefficients $(C_{\boldsymbol{w}}, r_{\boldsymbol{w}})$. Since the two velocity components are not correlated, the total ADSD is the sum of the ADSD of each velocity component. In the inertial range, this reads:

$$A(\kappa) = \underbrace{A_{\boldsymbol{w}}(\kappa)}_{\substack{\text{Leading at} \\ \text{large scales}}} + \underbrace{A_{\boldsymbol{\sigma}\dot{\boldsymbol{B}}}(\kappa)}_{\substack{\text{Leading at} \\ \text{small scales}}} \approx C_{\boldsymbol{w}} \kappa^{-r_{\boldsymbol{w}}}. \tag{5}$$

Therefore, the unresolved ADSD is chosen to compensate the resolved ADSD roll off – introduced by the deterministic subgrid parameterization – at small scales. Specifically, the unresolved ADSD is set to:

$$A_{\boldsymbol{\sigma}\dot{\boldsymbol{B}}}(\kappa) \quad = \quad \max(0, C_{\boldsymbol{w}} \kappa^{-r_{\boldsymbol{w}}} - A_{\boldsymbol{w}}(\kappa)) \, f_{BP}^2(\kappa). \tag{6}$$

As previously, $f_{BP}^2$ is a band-pass filter between $\kappa_m$ and $\kappa_M$. In practice, we set $\kappa_M$ to the theoretical resolution:

$$\kappa_M = \frac{\pi}{\Delta x}, \tag{7}$$

and $\kappa_m$ to the effective resolution which is estimated as follows:

$$\kappa_m \quad = \quad \left( \frac{\ln(0.95)}{\ln(0.1)} \right)^{1/p} \kappa_M, \tag{8}$$

where $p$ is the order of the Laplacian used as deterministic subgrid tensor (i.e. $\frac{Db}{Dt} = -\nu(-\Delta)^p b$). The justification of the above formula is left in the Appendix B. Compared to the work of Resseguier et al. (2017b), the value of $\kappa_m$ is less critical.

Indeed, Eq. (6) implies a weaker unresolved ADSD at larger scales where the resolved ADSD, $A_{\boldsymbol{w}}$ is stronger. This softens the threshold effect introduced by the band-pass filter $f_{BP}$.

In practice, we set an upper-bound for the estimation of $r_w$. Without this upper bound, a concentration of energy at relatively large wave-numbers – scales smaller than $\kappa_m$ – in the resolved fields can become unstable. Indeed, this localized energy concentration would decrease the $r_w$ estimation, and hence increase the unresolved ADSD $A_{\boldsymbol{\sigma}\dot{\boldsymbol{B}}}$ through (6) at large wave-

numbers – larger than $\kappa_m$. This implies a larger noise intake, which can induce a larger concentration of energy at relatively large wave-numbers in the resolved fields, resulting in a positive feedback loop. To prevent these unphysical instabilities, the slope $r_w$ is bounded.

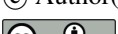



We have proposed a way to compute the unresolved ADSD $A_{\boldsymbol{\sigma}\dot{\boldsymbol{B}}}$ from the large-scale velocity statistics. From that ADSD,
it is possible to sample the unresolved velocity, as explained in the following.

A two-dimensional homogeneous divergence-free small-scale velocity can be constructed by filtering a one-dimensional
white noise $\dot{B}$ with a two-dimensional divergence-free filter $\breve{\boldsymbol{\sigma}} = \boldsymbol{\nabla}^{\perp}\breve{\psi}_{\sigma}$ (Resseguier et al., 2017b):

$$\boldsymbol{\sigma}\dot{\boldsymbol{B}} = \breve{\boldsymbol{\sigma}} \star \dot{B} = \boldsymbol{\nabla}^{\perp}\breve{\psi}_{\sigma} \star \dot{B}, \tag{9}$$

where $\star$ denotes a spatial convolution. This velocity field can be easily sampled in Fourier space:

$$\widehat{\boldsymbol{\sigma}\dot{\boldsymbol{B}}}(\boldsymbol{k}) = i\boldsymbol{k}^{\perp}\widehat{\breve{\psi}}_{\sigma}(\boldsymbol{k})\widehat{\dot{B}}(\boldsymbol{k}) = \frac{1}{\sqrt{\Delta t}}\, i\boldsymbol{k}^{\perp}\widehat{\breve{\psi}}_{\sigma}(\|\boldsymbol{k}\|)\, \frac{\widehat{\mathrm{d}B_t}}{\sqrt{\Delta t}}(\boldsymbol{k}), \tag{10}$$

where $\widehat{\mathrm{d}B_t}$ is the spatial Fourier transform of $\mathrm{d}B_t$, and $\frac{\mathrm{d}B_t}{\sqrt{\Delta t}}$, a discrete scalar white noise process of unit variance in space and
time. Thus, the small-scale velocity is defined by the Fourier transform of the streamfunction kernel, $\widehat{\breve{\psi}}_{\sigma}$.

In order to link the unresolved ADSD to the kernel $\breve{\boldsymbol{\sigma}}$ which defines the unresolved velocity, we note that

$$A_{\boldsymbol{\sigma}\dot{\boldsymbol{B}}}(\kappa) = E_{\boldsymbol{\sigma}\dot{\boldsymbol{B}}}(\kappa)\mathrm{d}t = \frac{1}{\mu(\Omega)} \oint_{[0,2\pi]} \mathrm{d}\theta_{\boldsymbol{k}}\kappa \left\| \widehat{\breve{\boldsymbol{\sigma}}}(t,\kappa) \right\|^2 = 2\pi\kappa^3 \left| \widehat{\breve{\psi}}_{\sigma}(t,\kappa) \right|^2, \tag{11}$$

where $\mu(\Omega)$ is the surface of the spatial domain $\Omega$ and $\theta_{\boldsymbol{k}}$ is the angle of the wave-vector $\boldsymbol{k}$. From formulas (6)-(11), we can
finally express the unresolved velocity as follow:

$$\begin{aligned}
\widehat{\boldsymbol{\sigma}\dot{\boldsymbol{B}}}(t,\boldsymbol{k}) &= \frac{1}{\sqrt{\Delta t}}\, i\boldsymbol{k}^{\perp}\widehat{\breve{\psi}}_{\sigma}(t,\|\boldsymbol{k}\|)\, \frac{\widehat{\mathrm{d}B_t}}{\sqrt{\Delta t}}(\boldsymbol{k}), \tag{12}\\[2mm]
&= \frac{1}{\sqrt{\Delta t}}\, i\boldsymbol{k}^{\perp}\sqrt{\frac{\max(0, C_{\boldsymbol{w}}\|\boldsymbol{k}\|^{-r_{\boldsymbol{w}}} - A_{\boldsymbol{w}}(\|\boldsymbol{k}\|))}{2\pi\|\boldsymbol{k}\|^3}}\, f_{BP}(\|\boldsymbol{k}\|)\, \frac{\widehat{\mathrm{d}B_t}}{\sqrt{\Delta t}}(\boldsymbol{k}). \tag{13}
\end{aligned}$$

Again the simulated unresolved velocity ADSD is physically relevant while the KE spectrum is not. Indeed, the simulated
unresolved velocity ADSD is expected to match the true (time-correlated) unresolved velocity ADSD, whereas the KE spectra
of the simulated and true unresolved velocities differ. Indeed, the true unresolved velocity correlation time spectral distribution
$\tau(\kappa)$ is not restricted to the time step $\mathrm{d}t$.

### 2.1.2 Heterogeneity

Our stochastic parameterization randomly folds tracer isolines. This process is often desirable. For instance, it can trigger
physically-relevant instabilities, such as the filament instabilities highlighted by Resseguier et al. (2017b). After these insta-
bilities have been randomly triggered, eddies are formed by non-linear processes. In a similar way, figures 2 and 3 show that
our stochastic dynamics enables a more realistic eddy distribution than deterministic simulations. However, a homogeneous
small-scale velocity may also perturb the tracer isolines which should remain still (i.e. which remain still in high-resolution
deterministic simulations), e.g. sharp, straight, coherent fronts. Figure 3 also highlights this drawback. A typical application
of this problem in more realistic flow simulations is the simulation of jets like the Gulf Stream and regions of the Antarctic
Circumpolar Current. These real-world jets are associated with diffusivity suppression (Ferrari and Nikurashin, 2010), and this

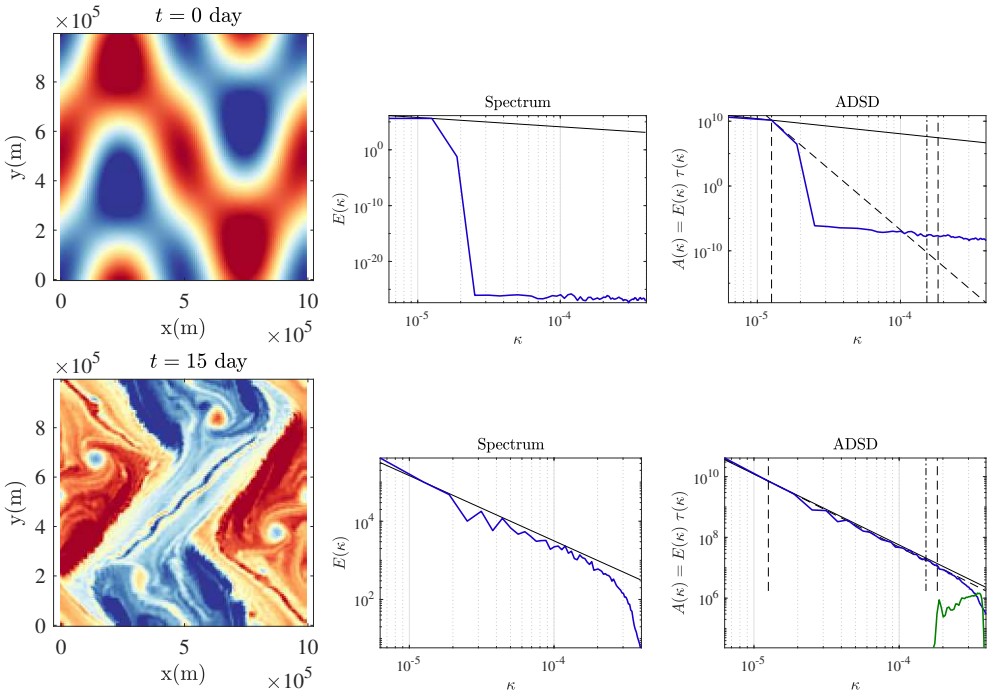

**Figure 1.** Buoyancy field ($m.s^{-2}$) (left), KE spectrum ($m^2.s^{-2}/(\text{rad}.m^{-1})$) (middle) and ADSD ($m^2.s^{-1}/(\text{rad}.m^{-1})$) (right) of the re-solved velocity, at $t = 0$ day (top) and $t = 15$ days (bottom), in blue, ADSD of the unresolved velocity (without a multiplicative constant), in green, and slope $-\frac{5}{3}$ (middle) and corresponding ADSD (right) in black solid line. The two dashed vertical lines define the interval where coefficients $C_w$ and $r_w$ are fitted. The right-hand-side dashed vertical line is fixed (see Appendix B for its specification). The left-hand-side dashed line corresponds to the energy-injecting scale. This scale is estimated – at each time step – as corresponding to the maximum of the compensated KE spectrum (i.e. KE spectrum divided by a theoretical spectrum). The dashed oblique line is the resulting fit (it is set to meet $A_w$ in the left bound of the fitting interval). The unresolved velocity is still restricted to a narrow spectral band, specifically between the dotted dashed line and the right end of the plot. Nevertheless, the form of its spectrum varies in time and is determined by the spectrum of the large-scale velocity. This particular initial condition corresponds to the case 2 studied by Constantin et al. (1994, 1999, 2012) at a resolution $128^2$.





effect is not present in our formulation so far. If we seek to preferentially perturb some tracer gradients, a heterogeneous small-scale velocity is required. Note that the heterogeneity discussed here needs to be non-stationary and thus cannot be represented by the stationary EOF presented later in this paper. Besides, in a small ensemble of realizations, relevant heterogeneity of the

small scales may make the spreading more accurate for UQ and enable comparable ensemble forecast accuracy with fewer members. We here propose a possible heterogeneous version of the previous method.

In order to obtain a heterogeneous model of the unresolved velocity, we need a heterogeneous version of the ADSD (4). Since the wave-number $\kappa$ cannot depends of the position $\boldsymbol{x}$, the constant $C_{\boldsymbol{w}}$ and/or the spectrum slope $r$ should do. A spatially varying spectrum slope is probably difficult to estimate. Hence, we restrict ourselves to a spatially-varying constant $C_{\boldsymbol{w}}$ and

a spatially-homogeneous spectrum slope. The constant may also varies with the time and the wave number. According to the Kolmogorov theory (e.g. Frisch, 1995) and (3):

$$C_{\boldsymbol{w}}(\boldsymbol{x},t,\kappa) = \mathrm{cst}.\epsilon_F^p(\boldsymbol{x},\kappa), \tag{14}$$

where $\epsilon_F$ is the energy flux through the spatial scales and $p = 1/3$ for a SQG flow. More specifically, the energy flux describes the energy moving from scales larges than $1/\kappa$ toward smaller scales and can be computed as follows:

$$\epsilon_F(\boldsymbol{x},t,\kappa) \overset{\triangle}{=} \overline{q_\kappa^<((\boldsymbol{w}\cdot\boldsymbol{\nabla})q)_\kappa^<}, \tag{15}$$

where $q$ is the transported (up to possible source terms) quantity, $g_\kappa^<$ is the low-pass filtered version of $g$ (setting to zero the Fourier modes of $g$ which have frequencies larger than $\kappa$) and $\overline{\bullet}$ stands for the spatial average (Frisch, 1995). For a SQG flow, $q$ corresponds to the buoyancy normalized by the stratification: $q = b/N$. The energy flux is essentially a third-order moment. It is very important because it describes the cascade of the flow by non-linear energy transfers (Frisch, 1995).

If the energy flux through scale is understood locally in space (as indeed Smagorinsky (1963) also assumes), the formula (14) provides a natural parameterization of the unresolved velocity heterogeneities. We simply modulates the unresolved ADSD (6) by the heterogeneous ratio $\epsilon_F^p/\overline{\epsilon_F^p}$ – averaged over the resolved inertial range wave-numbers:

$$\widetilde{\widehat{\boldsymbol{\sigma}\dot{\boldsymbol{B}}}}(t,\boldsymbol{k}) = \frac{1}{\sqrt{\Delta t}} i\boldsymbol{k}^\perp \sqrt{\frac{\max(0, C_{\boldsymbol{w}}\|\boldsymbol{k}\|^{-r_{\boldsymbol{w}}} - A_{\boldsymbol{w}}(\|\boldsymbol{k}\|))}{2\pi\|\boldsymbol{k}\|^3}} f_{BP}(\|\boldsymbol{k}\|) \frac{\widehat{\mathrm{d}B_t}}{\sqrt{\Delta t}}(\boldsymbol{k}), \tag{16}$$

$$\boldsymbol{\sigma}\dot{\boldsymbol{B}}(t,\boldsymbol{x}) = \mathcal{P}\bigg\{ \underbrace{\sqrt{\frac{\epsilon_F^p(t,\boldsymbol{x})}{\overline{\epsilon_F^p}(t)}}}_{\substack{\text{Heterogeneous}\\\text{modulation}}} \underbrace{\widetilde{\boldsymbol{\sigma}\dot{\boldsymbol{B}}}(t,\boldsymbol{x})}_{\substack{\text{Homogeneous}\\\text{velocity}}} \bigg\}, \tag{17}$$

where $\mathcal{P} = \mathbb{I}_d - \Delta^{-1}\boldsymbol{\nabla}\boldsymbol{\nabla}^T$ is the projector onto the space of free-divergence functions. This parameterization is physically meaningful since, locally in space, a stronger direct cascade at large scales (larger $\epsilon_F$ and thus larger $C_{\boldsymbol{w}}$) suggests that the unresolved velocity (large) should maintains this cascade by folding smaller-scale tracer structures. Furthermore, considering that the energy flux is a third-order structure makes this parameterization relevant to differentiate between strait fronts and curved structures (e.g. eddies). Indeed, at least three points are needed to define a curvature and differentiate between these

structures. Figure 3 confirms that this modulation enables a more accurate spatial distribution of the stochastic foldings.

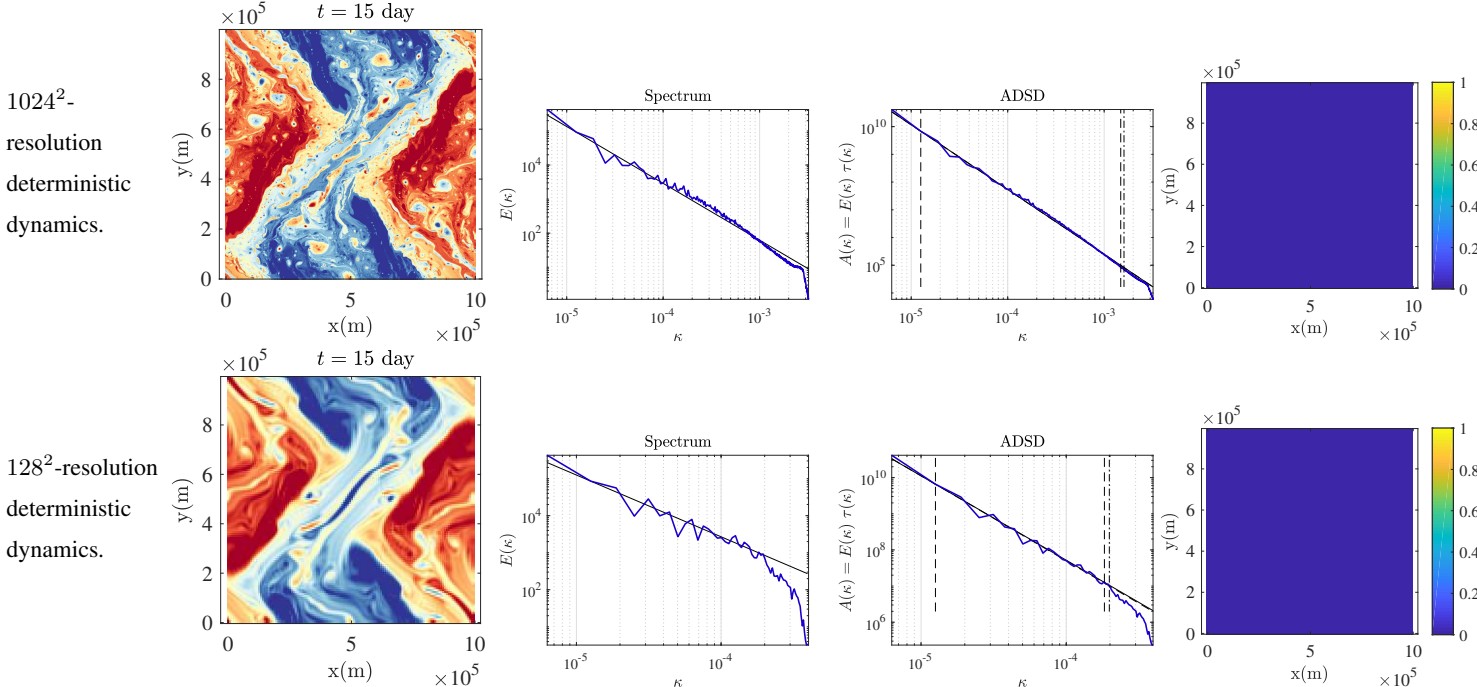

**Figure 2.** (From left to right) buoyancy field ($m.s^{-2}$), KE spectrum ($m^2.s^{-2}/(rad.m^{-1})$), ADSD ($m^2.s^{-1}/(rad.m^{-1})$) and variance tensor at $t = 15$ days for (from top to bottom) a $1024^2$-resolution deterministic dynamics and a $128^2$-resolution deterministic dynamics. This particular initial condition corresponds to the case 2 studied by Constantin et al. (1994, 1999, 2012) at a resolution $128^2$. The low-resolution deterministic dynamics shows too many filaments and not enough eddies.

In order to keep a divergence-free velocity and the ensuing properties (e.g. energy conservation), the modulated velocity is projected onto the space of free-divergence functions, using the operator $\mathcal{P}$. Because of that we do not consider the advection correction $\boldsymbol{w}^* - \boldsymbol{w} = -\frac{1}{2}(\boldsymbol{\nabla} \cdot \boldsymbol{a})^T$ of the LU formalism. Indeed, here the variance tensor has the simple form $\boldsymbol{a} = \frac{1}{d}\text{tr}(\boldsymbol{a})\mathbb{I}_d$. As such, the advection correction is a gradient field and is hence removed by the projection onto the space of free-divergence

functions.

Although relevant, the comparison of figures 2 and 3 – about front dynamics – remains qualitative. For a quantitative demonstration, adapted metrics should be used. Indeed, simple isotropic, homogeneous second-order statistics (e.g. KE spectra) cannot distinguish between eddies and filaments. Bi-spectra may overcome this drawback since they express three-point statistics. This quantitative analysis would necessitate studies of the metrics themselves. Therefore, these analyses will be addressed in

future work.

Many other closures rely on the Kolmogorov Kolmogorov (1941) model (3-14): in particular, the famous Smagorinsky model (Smagorinsky, 1963) and its variants (e.g. Fox-Kemper and Menemenlis, 2008; Bachman et al., 2017) provide a path to



**Figure 3.** (From left to right) buoyancy field $(m.s^{-2})$, KE spectrum $(m^2.s^{-2}/(rad.m^{-1}))$, ADSD $(m^2.s^{-1}/(rad.m^{-1}))$ and variance tensor at $t = 15$ days for $128^2$-resolution stochastic dynamics with (from top to bottom) homogeneous unresolved velocity , unresolved velocity modulated by $\|\boldsymbol{\nabla}b\|^{1/4}$, unresolved velocity modulated by $\epsilon_F^{1/6}$ . This particular initial condition corresponds to the case 2 studied by Constantin et al. (1994, 1999, 2012) at a resolution $128^2$. The low-resolution stochastic simulation do not show too many filaments and show enough eddies. Among the stochastic simulations, the energy-flux modulation better preserves the sharp straight fronts (e.g. front from $(0\ m, 2 \times 10^5\ m)$ to $(2.5 \times 10^5\ m, 5 \times 10^5\ m)$).





developing deterministic and dissipative scale-aware subgrid models. Typically, these models result in a Laplacian dissipation
which involves a heterogeneous eddy diffusivity or viscosity coefficient $\nu_{\mathrm{Sm}}$. Aside from their heuristic theoretical justification,
these Smagorinsky-type subgrid terms are formally equivalent to the turbulent dissipation of our stochastic model: $\boldsymbol{\nabla}\cdot\left(\frac{1}{2}\boldsymbol{a}\boldsymbol{\nabla}q\right)$
(Resseguier et al., 2017a) with $\boldsymbol{a} = 2\nu_{\mathrm{Sm}}\mathbb{I}_d$ and $q$ a transported quantity. This similarity suggests that a Smagorinsky-type model
could provide a good estimate for our variance tensor, and thus for the heterogeneity of the unresolved velocity.

The goal of the Smagorinsky model is to optimize the KE spectrum by targeting a specific turbulent diffusive scale adapted
to the simulation resolution. A turbulent dissipation coefficient expression can be derived from the Kolmogorov model (3,14)
and the closure:

$$\epsilon_F = \epsilon_D, \tag{18}$$

where $\epsilon_D$ is the dissipation: a second-order moment related to the molecular or turbulent diffusion. To develop a Smagorinsky-
type model, the resolved flux of energy-like conserved invariant is equated to the dissipation of the energy invariant:

$$\epsilon_D = \nu_{\mathrm{Sm}}\|\boldsymbol{\nabla}q\|^2. \tag{19}$$

From there, one can obtain an eddy diffusivity or viscosity coefficient proportional to $\|\boldsymbol{\nabla}q\|^h$. For a SQG flow, the exponent is
$h = 1/2$ where $q = b$ is the buoyancy (Bachman et al., 2017).

In the Kolmogorov theory of homogeneous and stationary turbulence, the energy flux is a constant of the flow and the
closure (19) is an exact result of a simple energy budget over an ensemble mean. Indeed, there is no accumulation of energy at
any scales in a stationary regime. This closure is very useful since the dissipation (19) is generally much simpler to compute
than the energy flux. Nevertheless, in every flow realization, the energy flux and the diffusion vary with space and time, and
they do not match each other locally. For instance, a strong straight front of a SQG flow involves a large dissipation but no
energy cascade because the velocity is aligned with front (see the definition (15)). Moreover, in any bounded, limited resolution
situation, the inertial cascade range is limited so the energy flux through scale varies with the wave number, especially outside
the inertial range.

The discrepancy between energy flux and dissipation is not so much of an issue for the Smagorinsky model because its
aim is the optimization of a second-order statistics at small scales. Unfortunately, this closure cannot be used to simplify the
modulation computation in our parametric random model. Indeed, the stochastic dynamics relies on processes – such as folding
– associated with higher-order statistics. Figure 3 (middle right) illustrates this statement. The following unresolved velocity
parameterization is used there:

$$\widehat{\widetilde{\boldsymbol{\sigma}\dot{\boldsymbol{B}}}}(t,\boldsymbol{k}) = \frac{1}{\sqrt{\Delta t}}i\boldsymbol{k}^\perp\sqrt{\frac{\max(0,C_{\boldsymbol{w}}\|\boldsymbol{k}\|^{-r_{\boldsymbol{w}}}-A_{\boldsymbol{w}}(\|\boldsymbol{k}\|))}{2\pi\|\boldsymbol{k}\|^3}}f_{BP}(\|\boldsymbol{k}\|)\frac{\widehat{\mathrm{d}B_t}}{\sqrt{\Delta t}}(\boldsymbol{k}), \tag{20}$$

$$\boldsymbol{\sigma}\dot{\boldsymbol{B}}(t,\boldsymbol{x}) = \mathcal{P}\left\{\underbrace{\frac{\|\boldsymbol{\nabla}b\|^{1/4}(t,\boldsymbol{x})}{\sqrt{\|\boldsymbol{\nabla}b\|^{1/2}(t)}}}_{\substack{\text{Heterogeneous}\\\text{modulation}}}\underbrace{\widetilde{\boldsymbol{\sigma}\dot{\boldsymbol{B}}}(t,\boldsymbol{x})}_{\substack{\text{Homogeneous}\\\text{velocity}}}\right\}. \tag{21}$$





Along sharp straight fronts, the dissipation will be larger. Accordingly, the associated modulation $\|\boldsymbol{\nabla}b\|^{1/4}$ enhances the stochastic folding where one would need it to weaken.

### 250  2.2  Data-driven model for the unresolved velocity

In this section, we detail a procedure proposed by Cotter et al. (2018b) for the estimation of the (weighted) EOFs, $(\boldsymbol{\xi}_k)_k$, involved in the unresolved velocity definition:

$$\boldsymbol{\sigma}\dot{\boldsymbol{B}} = \sum_k \boldsymbol{\xi}_k dW_k/\mathrm{d}t. \tag{22}$$

If we denotes $\lambda_k$ the $L^2$ norm of those EOFs, their normalized versions $(\boldsymbol{\xi}_k = \boldsymbol{\xi}_k/\lambda_k)_k$ are the eigenvectors of the self-adjoint
operator

$$\boldsymbol{f} \mapsto \int_\Omega \left[\boldsymbol{\sigma}(\boldsymbol{x})\boldsymbol{\sigma}^T(\boldsymbol{y})/\mathrm{d}t\right]\boldsymbol{f}(\boldsymbol{y})\mathrm{d}\boldsymbol{y}, \tag{23}$$

defined by the the small-scale velocity covariance $\left[\boldsymbol{\sigma}(\boldsymbol{x})\boldsymbol{\sigma}^T(\boldsymbol{y})/\mathrm{d}t\right]$. $(\lambda_k/\mathrm{d}t)_k$ is the set of eigenvalues of this operator.

The data-driven methods of Cotter et al. (2018b) relies on Lagrangian paths defined at two "resolutions". The first paragraph of this section defines these two types of Lagrangian paths. Then, we propose several ways to identify the time increments of
the infinite-dimensional Brownian motion $\boldsymbol{\sigma}\boldsymbol{B}_t$ from these Lagrangian paths. Preprocessing of the increments are needed in order to meet some structural assumptions. After this, we relate the increments covariance to the EOFs.

#### 2.2.1  Preliminary definitions

We introduce two types of velocity field:

– a high-resolution velocity $\boldsymbol{v}$ on a fine mesh-grid ($512^2$),

– a low-resolution velocity $\overline{\boldsymbol{v}}$ on a coarse mesh-grid ($64^2$). This velocity field is a spatially-low-pass-filtered version of $\boldsymbol{v}$.

Then, two types of Lagrangian path are defined:

– a "high-resolution flow", $\boldsymbol{X}_{ij}(t_0,t)$, defined by the high-resolution velocity $\boldsymbol{u}$:

$$\frac{\mathrm{d}\boldsymbol{X}_{ij}}{\mathrm{d}t}(t) = \boldsymbol{u}\left(t, \boldsymbol{X}_{ij}(t_0,t)\right) \text{ and } \boldsymbol{X}_{ij}(t_0,t_0) = \boldsymbol{x}_{ij}, \tag{24}$$

with $\boldsymbol{x}_{ij} = (i\Delta x, j\Delta y)$.

– a "low-resolution flow", $\overline{\boldsymbol{X}}_{ij}(t_0,t)$, defined by the low-resolution velocity $\overline{\boldsymbol{u}}$:

$$\frac{\mathrm{d}\overline{\boldsymbol{X}}_{ij}}{\mathrm{d}t}(t_0,t) = \overline{\boldsymbol{u}}\left(t, \overline{\boldsymbol{X}}_{ij}(t_0,t)\right) \text{ and } \overline{\boldsymbol{X}}_{ij}(t_0,t_0) = \boldsymbol{x}_{ij}. \tag{25}$$





### 2.2.2 Candidate for the increments realization

In order to estimate the EOFs, $(\boldsymbol{\xi}_i)_i$, involved in (A5), we assume that we can observe increments

$$\boldsymbol{\sigma}\Delta\boldsymbol{B}_{m\Delta T} \quad = \quad \boldsymbol{\sigma}\left(\boldsymbol{B}_{(m+1)\Delta T} - \boldsymbol{B}_{m\Delta T}\right) \tag{26}$$

We will interpret the following residual flow increments as a realisation of the above:

$$\widetilde{\Delta\boldsymbol{X}}_{ij}^m \quad \triangleq \quad \boldsymbol{X}_{ij}(m\Delta T,(m+1)\Delta T) - \overline{\boldsymbol{X}}_{ij}(m\Delta T,(m+1)\Delta T). \tag{27}$$

### 2.2.3 Preprocessing

The increments are supposed to be centered and divergence free (as $\boldsymbol{\nabla}\cdot\boldsymbol{\xi}_i = 0, \forall i$). Therefore, after computing the residual flow increments $\left(\widetilde{\Delta\boldsymbol{X}}_{ij}^m\right)_m$, they are centered:

$$\Delta\boldsymbol{X}'^m_{ij} \quad \triangleq \quad \widetilde{\Delta\boldsymbol{X}}_{ij}^m - \hat{\mathbb{E}}\{\widetilde{\Delta\boldsymbol{X}}_{ij}^m\}, \tag{28}$$

with the estimator

$$\hat{\mathbb{E}}\{\widetilde{\Delta\boldsymbol{X}}_{ij}^m\} \triangleq \frac{1}{N}\sum_{m=0}^{N-1}\widetilde{\Delta\boldsymbol{X}}_{ij}^m, \tag{29}$$

and projected onto the space of divergence-free functions:

$$\Delta\boldsymbol{X}_{ij}^m \triangleq \mathcal{P}\{\Delta\boldsymbol{X}'^m_{ij}\} \text{ with } \mathcal{P} = \mathbb{I}_d - \Delta^{-1}\boldsymbol{\nabla}\boldsymbol{\nabla}^T. \tag{30}$$

### 2.2.4 Covariance, quadratic covariation and EOF

Then, we can define the EOFs by an estimate of the spatial covariance of the residual flow increments (averaging over the time index $m$) :

$$\sum_k \boldsymbol{\xi}_k(\boldsymbol{x}_{ij})\,\boldsymbol{\xi}_k^{\ T}(\boldsymbol{x}_{pq}) \quad = \quad \boldsymbol{\sigma}(\boldsymbol{x}_{ij})\boldsymbol{\sigma}^T(\boldsymbol{x}_{pq}) \tag{31}$$

$$\approx \quad \frac{1}{(N-1)\Delta T}\sum_{m=0}^{N-1}\Delta\boldsymbol{X}_{ij}^m\left(\Delta\boldsymbol{X}_{pq}^m\right)^T, \tag{32}$$

In order to properly define the EOFs, we must add the following orthogonal constraint:

$$\int_\Omega \boldsymbol{\xi}_i(\boldsymbol{x})^T\boldsymbol{\xi}_j(\boldsymbol{x})\mathrm{d}\boldsymbol{x} = 0 \text{ if } i \neq j. \tag{33}$$

Finally, after estimating the $(\boldsymbol{\xi}_k)_k$ off-line, the ensemble forecast can generated on-line with the formula (22).





# 3 Numerical simulations and uncertainty quantification

## 3.1 Surface Quasi-Geostrophic (SQG) model

### 3.1.1 Deterministic SQG

This is a simplified model to describe the ocean surface dynamics at mesoscales (i.e. horizontal length scale of the order of $100$ km) (Blumen, 1978; Held et al., 1995; Lapeyre and Klein, 2006a; Constantin et al., 1994, 1999, 2012; Lapeyre, 2017). The buoyancy, $b$, is transported at the ocean surface by a horizontal velocity field:

$$\frac{Db}{Dt} = S, \tag{34}$$

where $D_t$ stands for the horizontal (deterministic or stochastic) material derivative and $S$ represents possible sources and sinks. As the potential vorticity is assumed to be zero in the fluid interior, the Fourier transform of the velocity streamfunction, $\hat{\psi}$, is related to the Fourier transform of the buoyancy, $\hat{b}$, by the SQG relationship:

$$\hat{\psi} = \frac{1}{N\|\boldsymbol{k}\|}\hat{b}, \tag{35}$$

where $N$ is the stratification and $\boldsymbol{k}$ the wave-vector.

### 3.1.2 Stochastic SQG

The LU and SALT versions of the SQG dynamics are formally similar to the deterministic model. However, the buoyancy transport (34) has to be understood in the stochastic sense (A4). Furthermore, the SQG relationship (35) must be interpreted with

$$\boldsymbol{w}^* = \boldsymbol{\nabla}^{\perp}\psi = \boldsymbol{\nabla}^{\perp}(-\Delta)^{-1/2}(b/N), \tag{36}$$

in the SALT context and $\boldsymbol{w} = \boldsymbol{\nabla}^{\perp}\psi = \boldsymbol{\nabla}^{\perp}(-\Delta)^{-1/2}(b/N)$ in the LU one. Besides, in the LU SQG, the advecting drift $\boldsymbol{w}^\star$ has to be divergence-free. we enforce this constraint by projecting the drift correction $(\boldsymbol{w}^* - \boldsymbol{w})$ onto the space of free-divergence functions. So, we have:

$$\boldsymbol{w}^* = \mathcal{P}(\boldsymbol{w}^* - \boldsymbol{w}) + \boldsymbol{w} = -\tfrac{1}{2}\mathcal{P}(\boldsymbol{\nabla}\cdot\boldsymbol{a})^T + \boldsymbol{\nabla}^{\perp}(-\Delta)^{-1/2}(b/N), \tag{37}$$

where $\mathcal{P} = \mathbb{I}_d - \Delta^{-1}\boldsymbol{\nabla}\boldsymbol{\nabla}^T$. Indeed, the SQG model is derived from a QG model with a transport of the buoyancy at the surface. Then, the potential vorticity (PV) is assumed to be $0$ inside the fluid. In the SALT framework, the PV reads $0 = PV_{slt} = \boldsymbol{\nabla}^{\perp}\cdot\boldsymbol{w}^* + f + (f_0/N)^2\partial_z^2\psi$, $\boldsymbol{w}^* = \boldsymbol{\nabla}^{\perp}\psi$ and $b = f_0\partial_z\psi$ ) and $\boldsymbol{w} = \boldsymbol{\nabla}^{\perp}\psi$. For the LU dynamics, $0 = PV_{lu} = \boldsymbol{\nabla}^{\perp}\cdot\boldsymbol{w} + f + (f_0/N)^2\partial_z^2\psi$, $\boldsymbol{w} = \boldsymbol{\nabla}^{\perp}\psi$ and $b = f_0\partial_z\psi$ ).

Nevertheless, the slight difference between the SQG SALT and the SQG LU models are not considered in this section since we neglect the advection correction of the LU framework $(\boldsymbol{w}^* - \boldsymbol{w})$. So, we simulate the stochastic transport (34) coupled with the SQG relation (36). The unresolved velocity statistics encoded in $\boldsymbol{\sigma}$ are specified either by the self-similar method of Sect. 2.1.1 or by the data-driven method of Sect. 2.2.





### 3.1.3 Flow simulation

We perform a high-resolution simulation ($512^2$ spatial grid) of the deterministic SQG model with the following initial condition:

$$325 \quad b(\boldsymbol{x}, t = 0) = 0.2 B_0 \cos\left(\frac{2\pi}{L}(x + y)\right) \tag{38}$$
$$+ F\left(\boldsymbol{x} - \boldsymbol{x}_{00}^0\right) + F\left(\boldsymbol{x} - \boldsymbol{x}_{10}^0\right) - F\left(\boldsymbol{x} - \boldsymbol{x}_{01}^0\right) - F\left(\boldsymbol{x} - \boldsymbol{x}_{11}^0\right),$$

$$\text{with } F(\boldsymbol{x}) = B_0 \exp\left(-\frac{1}{2}\left(\frac{15}{L}\right)^2 \left(x^2 + \left(\frac{y}{2}\right)^2\right)\right), \tag{39}$$

$$\text{and } \boldsymbol{x}_{ij}^0 = \frac{L}{4}\begin{pmatrix} 1 \\ 1 \end{pmatrix} + \frac{L}{2}\begin{pmatrix} i \\ j \end{pmatrix}. \tag{40}$$

The source term, $S$, involves a hyperviscosity, a linear drag and an additive stationary forcing:

$$330 \quad S \triangleq -\nu_{\text{HV}}\Delta^4 b - \frac{1}{\tau_F}b + A_F \sin(k_F^x x)\sin(k_F^y y). \tag{41}$$

The parameters of the simulations are summed up in the table 1. The influence of the initial condition remains during about a month. Then, the turbulence is maintained by the forcing as illustrated by figure 4.

From $t = 50$ days to $t = 100$ days, the EOFs of the data-driven method are learned. From $t = 100$ days to $t = 130$ days, the deterministic high-resolution simulation is used as a reference. In this time interval, several stochastic low-resolution ($64^2$)

335 simulations are performed and compared. These simulations are initialized at $t = 100$ days with the reference high-resolution simulation projected at low resolution (i.e. keeping only the Fourier modes associated with coarse resolution grid).

**Table 1.** Parameters of the simulation

| Parameters | Value |
|---|---|
| $L$ (domain length) | $10^6\ m$ |
| $B_0/N$ | $3.24\ m.s^{-1}$ |
| $\sqrt{\overline{b^2}}(t = 0)/N$ | $1.16\ m.s^{-1}$ |
| $\sqrt{\overline{b^2}}(t = 100\text{days})/N$ | $2.28\ m.s^{-1}$ |
| $A_F \tau_F/N$ | $32.4\ m.s^{-1}$ |
| $1/\tau_F$ | $2.16 \times 10^{-7}\ s^{-1} \approx 1/(54\ \text{days})$ |
| $(k_F^x, k_F^y)$ | $(3(2\pi/L), 2(2\pi/L))$ |
| $\nu_{\text{HV}}/dx^8$ | $3.39 \times 10^{-8}\ s^{-1}$ |





**Figure 4.** Buoyancy ($m.s^{-2}$) at $t = 0, 10, 30, 50$ and $70$ days of advection (top) and its spectrum ($m^2.s^{-4}/(rad.m^{-1})$) at $t = 50$ days of advection (bottom) for the deterministic SQG model at resolution $512^2$.





## 3.2 Learning and analysis of the EOF

Here, we describe the convergence of the EOF estimation. The accuracy of this estimation is in particular a function of the number of snapshots used and of the Lagrangian advection time $\Delta T$. We first describe the convergence with the number of 340 snapshots and then – as a remark – the convergence with the Lagrangian advection time $\Delta T$.

The EOFs are learned between $t = 50$ days and $t = 100$ days from $12465$ spatial fields of residual flow increments. Even though a large number of snapshots are used, the correlation time of the residual flow increments is about $1$ day. This is not negligible compared to the estimation time window : $50$ days. Accordingly, the EOFs are not fully converged. But, we expect this convergence to be sufficient for the present work. Moreover, for real applications at this spatial scale, we expect 345 the unresolved velocity statistics to be non-stationary on temporal scales larger $50$ days. Thus, learning a unresolved velocity stationary statistical representation on larger time window might be difficult in practice. Therefore, even if our EOF estimation is not converged, it represents a realistic test case.

**Remark 1.** *The integration time of the flows $\Delta T$ is a critical parameter for the definition of the EOF. Indeed, for small advection time $\Delta T$, the length of an increment of a Lagrangian flow path is proportional to the velocity and to the advection* 350 *time. This is the so-called ballistic regime (Falkovich et al., 2001; Vallis, 2006). In particular, residual flow increments squared norm would be proportional to $\Delta T^2$, the variance tensor estimator proportional to $\Delta T$, and the estimated EOFs proportional to $\sqrt{\Delta T}$. With a larger advection time $\Delta T$, the Lagrangian velocity decorrelates – along the flow path – from the initial Lagrangian velocity. When the advection time becomes larger than the correlation time of the Lagrangian velocity, the flow path begins to act as a Brownian motion (Falkovich et al., 2001; Penland, 2003; Klyatskin, 2005; Vallis, 2006; Keating et al.,* 355 *2011). The length of a displacement scales as $\sqrt{\Delta T}$ i.e. the Lagrangian velocity acts as a white noise in time. This is the so-called diffusive regime. Figure 5 illustrates this convergence with the average tensor $a_0 = \frac{1}{d}\overline{\mathrm{tr}(\boldsymbol{a})}$ with $\Delta T$. In this paper, the Lagrangian advection time $\Delta T$ is computed from a CFL at the coarse resolution $64^2$ – about $300s$. Although it corresponds to the ballistic regime (i.e. the flow increments are correlated), this choice is coherent with the work of Cotter et al. (2018b) and gives very good UQ results. Moreover, the residual flow increments – and hence the EOFs – are spatially aliased since* 360 *a large part of those increments lives at small spatial scales but are spatially sampled on a coarse spatial grid $\boldsymbol{x}_{ij}$. Figure 6 confirms this idea. If all the $2 \times 64^2 - 1 = 8191$ EOFs are considered, the ADSD of the unresolved velocity reveals a strong spatial aliasing.*

Both the data-driven and the non-data driven methods exhibit large unresolved velocities at the smallest scales of the coarse resolution grid (see figure 6). Nevertheless, the two ADSD are distinct. In particular, the data-driven method involves some 365 large spatial scales. One could think that these large-scale components would disappear if the flow $\boldsymbol{X}$ and $\overline{\boldsymbol{X}}$ are integrated in a Eulerian way rather than in a Lagrangian way. However, the advection time $\Delta T$ being very small, few differences are expected between a Eulerian and a Lagrangian advection. New numeric experiments confirm this idea (not shown). A complete study of these effects is beyond the scope of this paper.





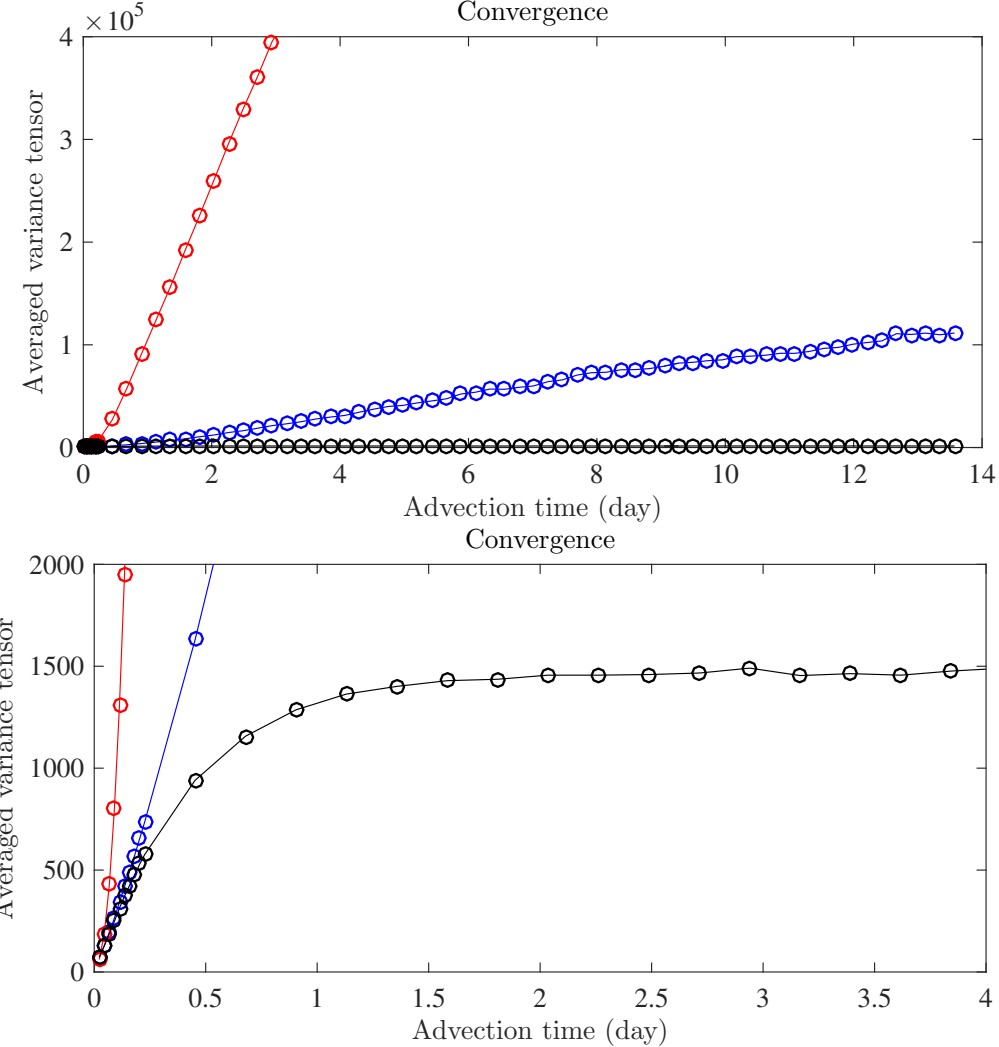

**Figure 5.** Estimation of the spatial average of the trace of the variance tensor divided by $d = 2$, $\frac{1}{Md} \sum_{q=1}^{M} \mathrm{tr}(\boldsymbol{a}(\boldsymbol{x}_q)) = \frac{1}{Md} \sum_{q,i} \|\boldsymbol{\xi}_i(\boldsymbol{x}_q)\|^2$, as a function of the Lagrangian advection time $\Delta T$, for the method 1 (described in Sect. 2.2) (blue plot) and for the method 2 (the flow increments defined by the integration of the small-scale velocity along the high-resolution flow : $\widetilde{\Delta \boldsymbol{X}}_{ij}^m = \int_{k\Delta T}^{(k+1)\Delta T} (\boldsymbol{u} - \overline{\boldsymbol{u}})(t, \boldsymbol{X}_{ij}(m\Delta T, t))\mathrm{d}t$ ) (black plot). The red plot relies on another method not described here. The bottom plot is a zoom of the top plot. For low $\Delta T$, the Lagrangian velocities remain highly correlated during the interval $[0, \Delta T]$, i.e. during the Lagrangian advection. It is the ballistic regime where the flow increments scale linearly in time: $\|\widetilde{\Delta \boldsymbol{X}}_{ij}^m\| \propto \Delta T$. In that regime, the methods 1 and 2 are very similar, since the high-resolution $\boldsymbol{X}_{ij}(m\Delta T, t)$ and low-resolution flows $\overline{\boldsymbol{X}}_{ij}(m\Delta T, t)$ remain close to each other. Above $\Delta T = 1$ day, the final and initial small-scale Lagrangian velocities, $(\boldsymbol{u} - \overline{\boldsymbol{u}})$, are decorrelated. It is the diffusive regime where the flow increments scale as the square-root of time: $\|\widetilde{\Delta \boldsymbol{X}}_{ij}^m\| \propto \sqrt{\Delta T}$. This is the relevant regime for the estimation of the EOF since we meet the fundamental assumption of our model: a Brownian behavior for the small-scale flow. For the method 1, the diffusive regime is not visible – at least in this advection time window – and the EOF estimation is theoretically not possible. In contrast, the method 2 provides a converged estimator of the variance tensor for an advection time $\Delta T > 1$ day.

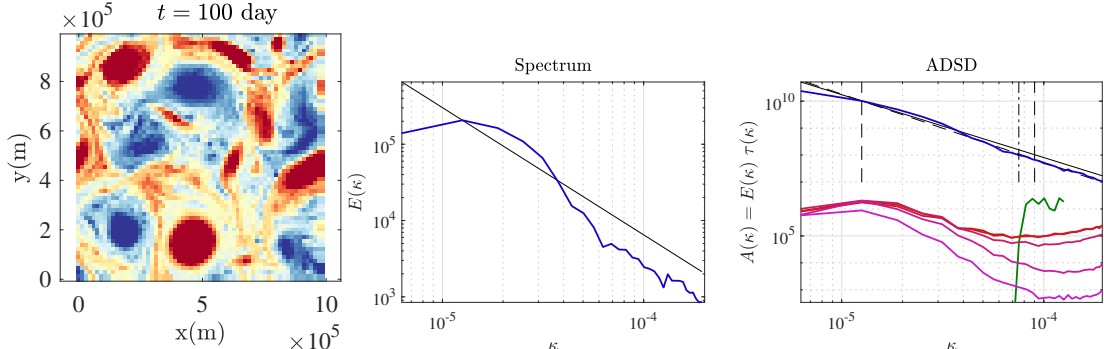

**Figure 6.** Buoyancy field ($m.s^{-2}$) (left), kinetic energy spectrum ($m^2.s^{-2}/(rad.m^{-1})$) (middle) and ADSD ($m^2.s^{-1}/(rad.m^{-1})$) (right) of $\boldsymbol{w}$, at $t = 17$ days, in blue, ADSD of $\boldsymbol{\sigma}\dot{\boldsymbol{B}}$ from the non-data-driven method (without a multiplicative constant), in green, ADSD of $\boldsymbol{\sigma}\dot{\boldsymbol{B}}$ from the data-driven method, with 2, 20, 200, 2000 and 8000 EOFs, in colors from pink to red, and slope $-\frac{5}{3}$ (middle) and corresponding ADSD (right) in black solid line. The two dashed vertical lines define the interval where coefficients $C_{\boldsymbol{w}}$ and $r_{\boldsymbol{w}}$ are fitted. The dashed oblique line is the resulting fit (it is set to match $A_{\boldsymbol{w}}$ in the left bound of the interval).

### 3.3 One realization

We now simulate the SALT-LU SQG dynamics (34)-(36), at low resolution ($64^2$), with two possible parameterizations for the unresolved velocity : either the data-driven model (see Sect. 2.2) or the parametric and self-similar model (see Sect. 2.1). For the data-driven model, we keep 200 EOFs. This choice will be explained in the next section. For all simulations, there is a unique reference initial condition at $t = 100$ days. This initial condition is the low-resolution ($64^2$) projection of the reference deterministic high-resolution ($512^2$) simulation (i.e. the Fourier modes associated with large wave vectors are set to zero and

the obtained spatial field is subsampled at the low resolution). Spatial fields, KE spectra and ADSD are plotted in figures 7 and 8. For comparison purposes, we also show the low-resolution deterministic SQG simulations and the low-resolution ($64^2$) projection of the reference deterministic high-resolution simulation. As already pointed out by Resseguier et al. (2017b) for free-decaying turbulence, one realization of the SALT-LU dynamics is – at least – not worse than a low-resolution deterministic simulation. At $t = 110$ days, all the spatial fields of figure 7 are still similar whereas at $t = 120$ days, the spatial fields strongly

differ. It probably means that the predictability time is about 2 weeks.

### 3.4 Uncertainty quantification

We now forecast two ensembles of 200 realizations following the stochastic SQG dynamics (34)-(36), with the same unique reference initial condition at $t = 100$ days. Again, the ensembles members evolve at low resolution ($64^2$). The first ensemble is generated with the data-driven model for the unresolved velocity (see Sect. 2.2) while the second ensemble is generated with

the parametric and self-similar model for the unresolved velocity (see Sect. 2.1). For the data-driven model, we keep only 200 EOFs since the number of EOFs cannot be larger than the ensemble size without increasing the complexity of the algorithm.

**Figure 7.** Buoyancy fields $(m.s^{-2})$ (left), kinetic energy spectra $(m^2.s^{-2}/(\mathrm{rad}.m^{-1}))$ (middle) and ADSDs $(m^2.s^{-1}/(\mathrm{rad}.m^{-1}))$ (right) of $\boldsymbol{w}$, in blue, ADSDs of $\boldsymbol{\sigma\dot{B}}$, in green, at $t = 110$ days, for (from top to bottom) the low-resolution $(64^2)$ projection of the reference deterministic high-resolution $(512^2)$ SQG dynamics ; the low-resolution $(64^2)$ deterministic SQG dynamic ; one realization of the low-resolution $(64^2)$ SALT-LU SQG dynamics with self-similar parameterization; one realization of the low-resolution $(64^2)$ SALT-LU SQG dynamics with data-driven parameterization.



Low-resolution ($64^2$) projection of the reference deterministic high-resolution ($512^2$) SQG dynamics.

Low-resolution ($64^2$) deterministic SQG dynamic.

One realization of the low-resolution ($64^2$) SALT-LU SQG dynamics with self-similar parameterization.

One realization of the low-resolution ($64^2$) SALT-LU SQG dynamics with data-driven parameterization.

**Figure 8.** Buoyancy fields ($m.s^{-2}$) (left), kinetic energy spectra ($m^2.s^{-2}/(\mathrm{rad}.m^{-1})$) (middle) and ADSDs ($m^2.s^{-1}/(\mathrm{rad}.m^{-1})$) (right) of $\boldsymbol{w}$, in blue, ADSDs of $\boldsymbol{\sigma \dot{B}}$, in green, at $t = 120$ days, for (from top to bottom) the low-resolution ($64^2$) projection of the reference deterministic high-resolution ($512^2$) SQG dynamics ; the low-resolution ($64^2$) deterministic SQG dynamic ; one realization of the low-resolution ($64^2$) SALT-LU SQG dynamics with self-similar parameterization; one realization of the low-resolution ($64^2$) SALT-LU SQG dynamics with data-driven parameterization.





With such ensemble forecasts, we aim at representing the variety of possible behaviors of the fluid dynamics system. In particular, the spreading (i.e. the variance increase) of an ensemble is expected to make an ensemble be closer to the reference. In such a case, the standard deviation – at each point and at each time – is expected to be of the order of the bias. Figure 9
shows that both the data-driven and the self-similar parameterizations achieve this goal everywhere and for every time. The point-wise biases of the data-driven method, of the self-similar method and of a deterministic simulation (at the same low resolution) are very similar (not shown). Therefore, we only plot the point-wise bias of the data-driven method. In figure 9, the represented biases and error estimations (1.96 times the point-wise standard deviation) are normalized by the squared energy of the reference solution. Note that those relative point-wise biases increase very quickly with time.

After $t = 105$ days and especially after $t = 115$ days, a bifurcation plays a large role in the simulation error and its estimation. This bifurcation is due to the chaotic trajectory of an eddy – centered on $(525\text{km}, 300\text{km})$ at $t = 105$ days. Due to the incorrect trajectory in the ensemble mean, a large yellow spot develops in the bias images. Both ensembles capture well this variability by creating similar spots. According to the doubling of those spots, there are probably only two likely trajectories for this eddy, at least until $t = 118$ days.

Similar results UQ results have been obtained on a free turbulence flow (Resseguier et al., 2017b). This close analysis has also compared the UQ potential of LU/SALT algorithms against that of a deterministic dynamics with random initial conditions. The latter has shown an underestimation of errors by one order of magnitude.

Figure 9 offers a visual validation of our methods' UQ potential in the whole spatial domain. Nevertheless, the point-wise laws of the ensembles are not Gaussian, since we consider a non-linear evolution law with multiplicative noise. Therefore, the
point-wise confidence interval is not in general a symmetric interval centered on the point-wise mean with width $2 \times 1.96$ times the point-wise standard deviation. Such an interval is only a first approximation. In order to be more accurate in our analysis, we now focus on few spatial points. There, we compute – at each time – the true ensemble-based confidence intervals at $95\%$ and at $50\%$ and compare them with the reference value. Figure 10 display the results at 2 grid points. Again, the results are impressively good. Similar UQ proxies have been presented by Cotter et al. (2018b) with a SALT 2D Euler dynamics with
Dirichlet boundary conditions. This work also highlights the excellent UQ skills of our frameworks.

To conclude this quantitative UQ analysis, we study the effect of the number of realizations needed. How many is a main issue in operational weather forecast centers since each realization is costly. Accordingly, we start again the UQ analysis but with 20 ensemble members only. Figure 11 shows that the $97.5\%$ and the $2.5\%$ quantiles get closer, compared to the 200-ensemble-member case. It means that the probability density tail estimations – i.e. the representations of extremes – are slower
to converge than the more likely values. Nevertheless, the ensembles still capture well the reference dynamics of the center of the distribution.

## 4   Conclusion

This paper develops the SALT and LU models which coexist in a single family of stochastic schemes. In addition to their general theoretical properties, we have discussed and compared possible parameterizations for this new scheme family.


**Figure 9.** Normalized buoyancy bias absolute value, $|\widehat{\mathbb{E}}\{b\} - b^{ref}|$, (dimensionless) of the stochastic SQG model (left) and its estimation (1.96× the standard deviation of the ensemble) for the data-driven method (middle) and the self-similar one (right) at resolution $64^2$ at (from top to bottom) at $t = 105, 110, 115$ and $120$ days. The reference is the usual SQG model at resolution $512^2$ – adequately filtered and subsampled. The stochastic simulations and the reference have a common initial condition at $t = 100$ days.





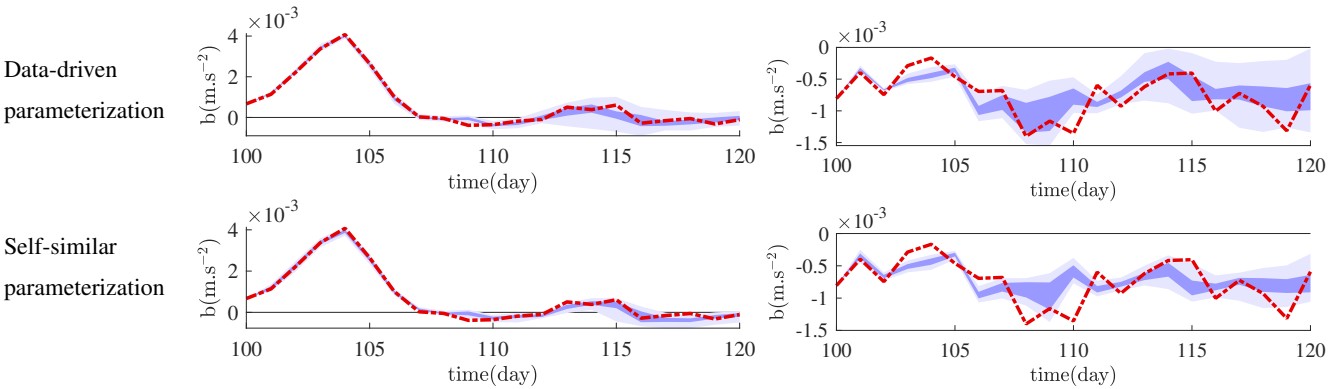

**Figure 10.** Confidence interval at 95% (light purple) and at 50% (dark purple) along time, computed from the low-resolution stochastic SQG 200-member-ensemble with the data-driven (top) and with the self-similar (bottom) parameterization, at the points (500km,250km) (left) and (250km,500km) (right). The high-resolution reference is superimposed in red.

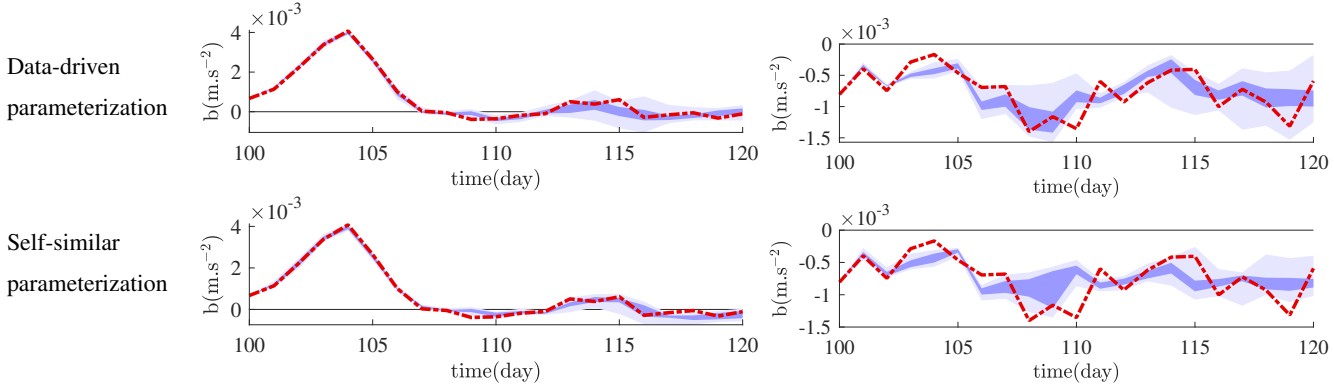

**Figure 11.** Confidence interval at 95% (light purple) and at 50% (dark purple) along time, computed from the low-resolution stochastic SQG 20-member-ensemble with the data-driven (top) and with the self-similar (bottom) parameterization, at the points (500km,250km) (left) and (250km,500km) (right). The high-resolution reference is superimposed in red.





The appendix A highlights the strong theoretical similarities between SALT and LU stochastic subgrid tensors at the heart of their parameterizations. These frameworks assume that tracers are transported by the sum of a resolved large-scale velocity and an unresolved time-uncorrelated velocity. As already mentioned in the literature, these subgrid models can conserve some but not all invariants. The SALT framework imposes helicity and circulation conservation in two and three dimensions and enstrophy conservation in two dimensions, whereas LU dynamics strictly conserves kinetic energy. Yet, for a homogeneous unresolved turbulence, we have proved that – on average – LU dynamics also conserve the helicity and circulation.

This paper mainly focuses on numerical parameterization. We have formulated and described several parameterizations which apply to both SALT and LU frameworks. A SALT-LU parameterization means choosing a spatial covariance for the unresolved small-scale velocity (i.e. choosing $\sigma$). The stationary homogeneous self-similar parameterization of Resseguier et al. (2017b) has been improved to make it non-stationary and tuning-free. Spectral properties are learned "on the fly" from the resolved large-scale velocity as is common in Large Eddy Simulation practice. A heterogeneous modulation of that parameterization – based on the energy flux through scales – is also proposed. This modulation naturally comes into play when considering Kolmogorov-like energy cascades and spectra. Because the energy flux quantifies the energy cascade – which is assumed to be constant across scales – the modulated unresolved velocity acts only where there is a large-scale energy cascade. This parameterization improvement enables a better simulation of straight strong fronts in SQG dynamics. However, the usual convenient approximation produced in the Smagorinsky diagnosis of the energy flux – the dissipation – cannot be used accurately here. We also recall the stationary heterogeneous data-driven parameterization method of Cotter et al. (2018b, a). Here, small Lagrangian displacement increments are computed at distinct resolutions from high-resolution simulation outputs. The spectral decomposition of the Lagrangian displacements covariance leads to a light and convenient representation of the unresolved velocity statistics: the Lagrangian displacements EOFs.

Finally, two tuning-free SALT-LU parameterizations have been numerically compared: the new non-stationary homogeneous self-similar parameterization and the stationary heterogeneous data-driven one of Cotter et al. (2018b, a). The test case is a homogeneous and stationary forced SQG turbulence. A single realization of a SALT-LU simulation is found to be – at least – as good as the result of the corresponding deterministic simulation at the same resolution. For both parameterizations, SALT-LU ensemble forecasts are found to predict the right amplitudes and positions of numerical errors. The uncertainty skills of SALT-LU ensemble forecasts remain impressively good even with only 20 realizations.

## 5 Discussion

A lot of theoretical and practical questions remain about SALT-LU schemes.

The numerical explorations of this paper have focused on SQG dynamics because – except for the neglected advection correction – SALT and LU SQG models coincide. This simplification has enabled a clearer parameterization comparison. An interesting dual study would be the comparison of distinct SALT and LU models, but with a fixed parameterization (i.e. fixed $\sigma$ model).





For this purpose, the simplest dynamics to consider would be two-dimensional incompressible Navier-Stokes equations. The three-dimensional or Quasi-Geostrophic (QG) dynamics would be appropriate although possess a larger parameter space and more costly computation. The SALT version will conserve helicity and circulation while the LU one will conserve kinetic

energy. It would be a very interesting exercise to examing the inertial cascades of energy, enstrophy, and potential enstrophy in these systems and how they are affected by the SALT vs. LU assumptions. Nonetheless, even an understanding of these cascades is probably not sufficient to objectively conclude which framework is more appropriate in general, as the answer will likely depend on the application, model resolution, etc. A variety of idealized and applied numerical studies will probably be necessary to gain intuition for how to optimize in a varity of settings. And even for one specific situation, the quality and skill

metrics of choice are not obvious. Even in the simple cases studied here, the usual second order statistics are probably not sufficient to discriminate between these dynamics. Furthermore, it is important to note that the exact conservation of vorticity by the SALT scheme and energy by the LU scheme may be counterproductive in certain heterogeneous settings where the small-scale features are required to exhibit systematic property transport. One interesting example is the oceanic wind-driven gyre, whose circulation magnitude depends critically on the gyre and its turbulence to relocate the source of vorticity and

energy from the wind to the regions where dissipation occurs. An eddy vorticity transport across large-scale streamlines (i.e., affecting the interpretation of the Kelvin circulation theorem) is needed Fox-Kemper and Pedlosky (2004). A cross-streamline energy transport is also part of the system equilibration Scott and Straub (1998). In the real ocean, the eddies shed in the Aghulas retroflection are a classic example of organized small-scale transport Gordon (1985) as are the "eddy cannons" that fire mesoscale eddies across the Antarctic Circumpolar Current Hallberg and Gnanadesikan (2006).

In the presence of turbulence heterogeneity, another distinction between SALT and LU models is the advection correction. It is also the single distinction between LU dynamics and Mikulevicius and Rozovskii (2004a). So far, it is not clear what constitutes a "large-scale velocity" in practice. Again, this probably depends on the situation. Even though Cotter et al. (2017) have provided some first theoretical clues, further explorations are needed. Numerical and experimental work will probably help if one identifies and studies simple heterogeneous flows meeting the main assumption of SALT-LU models : the velocity

time-scale gap.

The SALT-LU parameterizations presented in this study (e.g. the ADSD method) could be naturally extended to physical-domain-based implementation, as is an important step in evaluating schemes for operational use, e.g., Pearson et al. (2017). Indeed, Sect. 2.1 details this tuning-free parameterization in the Fourier space only. A convenient dual of self-similar spectra

(i.e. spectra scaling as $\|\boldsymbol{k}\|^{\alpha}$) in the physical space is the widely-used Matérn covariance (Williams and Rasmussen, 2006; Lim and Teo, 2009; Lilly et al., 2017). Thus, the spatial filter $\breve{\boldsymbol{\sigma}}$ – appearing in $\boldsymbol{\sigma} \mathrm{d}\boldsymbol{B}_t = \breve{\boldsymbol{\sigma}} * \mathrm{d}\boldsymbol{B}_t$ – of a possible future two-dimensional physical-domain-based ADSD parameterization would probably be built from the curl of a Matérn covariance.

Nevertheless, the homogeneity of the ADSD parameterization is a drawback. Heterogeneous solutions need to be developed. Unfortunately, the ADSD heterogeneous modulation of this paper remains expensive to compute, even making use of the

Fourier space. Moreover, even though its third-order physical meaning and its natural association with SALT-LU is very instructive, its additive value for UQ is not clear.


The heterogeneous parameterization method of Cotter et al. (2018b, a) is valuable in the presence of some turbulence heterogeneities (e.g. heterogeneities induced by fixed boundary conditions). However, the practical usefulness of this parameterization remains debatable, because its heterogeneity is assumed to be stationary. Other ways for calibrating the subgrid unresolved statistics using machine learning is also on going research.

Many SALT-LU parameterizations are now available. Regardless of their differences, the resulting stochastic dynamics and associated UQ skill seems to be relatively independent of the particular parameterization choice. Notwithstanding, we think that learning-free heterogeneous and non-stationary parameterizations may further improve SALT-LU UQ skills and/or enable smaller ensemble size and thus more efficient computation.

Finally, the main goal of all these studies is SALT-LU-based data assimilation. Cotter et al. (2019) have opened the way, using a particle filter and the heterogeneous parameterization method of Cotter et al. (2018b, a). Observed data from a dynamical system of $O(10^6)$ degrees of freedom were successfully assimilated into the parameterized system of $O(10^4)$ degrees of freedom. We expect that many other data assimilation studies will follow.

**Author contribution**

V. Resseguier and W. Pan decided to perform this SALT-LU numerical study. B. Fox-Kemper obtained the funding and supervised the project. V. Resseguier and B. Fox-Kemper designed the ADSD methodology and its heterogeneous variants. V. Resseguier developed the code and perform the simulations. W. Pan helped to reproduce the data-driven parameterization of Cotter et al. (2018b). All authors prepared the manuscript.

**Acknowledgments**

We would like to warmly thank Darryl D. Holm, Dan Crisan, Colin Cotter, Igor Shevchenko, Bertrand Chapron and Etienne Mémin for helpful discussions and for having enabled the collaborations which have lead to this paper.

**Competing interests statement**

One of the author is currently employed by a private company named "SCALIAN".

**Funding**

Two authors were partially funded by Brown University and by NSF OCE 1350795 and ONR N00014-17-1-2963. Another author was funded by the Engineering and Physical Sciences Research Council (EPSRC) grant EP/N023781/1.





## Appendix A: Theoretical motivations

Here, we briefly highlight the similarities and differences between the location uncertainty (LU) model and the stochastic Lie
transport (SALT) model. To simplify the comparison, we work in the Stratonovich representation and restrict ourselves to the
incompressible case.

### A1  Lagrangian path

From a large-scale under-resolved point of view, SALT and LU methods assume that the fluid velocity is partially uncorrelated
in time. Hence, the position of a Lagrangian particle $\boldsymbol{X}_t$ evolves in time according to:

$$\frac{\mathrm{d}\boldsymbol{X}_t}{\mathrm{d}t} = \boldsymbol{w}^*(\boldsymbol{X}_t,t) + \left(\boldsymbol{\sigma}\dot{\boldsymbol{B}}\right)(\boldsymbol{X}_t,t), \tag{A1}$$

where $\boldsymbol{\sigma}\dot{\boldsymbol{B}}$ is time-uncorrelated and $\boldsymbol{w}^*$ is the Stratonovich large scale drift term. Formally, for a spatial domain $\Omega \subset \mathbb{R}^d$, the
process $(t \mapsto \boldsymbol{B}_t)$ is a cylindrical $\mathbb{I}_{d(\mathcal{L}^2(\Omega))^d}$-Wiener process (see Da Prato and Zabczyk (1992) and Prévôt and Röckner (2007)
for more information on infinite-dimensional Wiener processes and cylindrical $I_d$-Wiener processes). Recently, Cotter et al.
(2017) have rigorously shown that such a decomposition corresponds to the limit of a deterministic flow when the correlation
time of the small-scale velocity goes to zero.

### A2  Notation correspondences

The Lagrangian equation can also be written with Ito notations:

$$\boldsymbol{X}_{t+\mathrm{d}t} - \boldsymbol{X}_t = \boldsymbol{w}(\boldsymbol{X}_t,t)\mathrm{d}t + \boldsymbol{\sigma}(\boldsymbol{X}_t,t)(\boldsymbol{B}_{t+\mathrm{d}t} - \boldsymbol{B}_t). \tag{A2}$$

Note the difference in the drift term when compared with (A1). Table A1 summarizes the notations differences between SALT
and LU both for Ito and Stratonovich notations.

### A3  Scalar transport

For a – possibly active – scalar tracer denoted $q$, the SALT prescribes the same type of evolution equation as the LU models:

$$\frac{Dq}{Dt}(\boldsymbol{x},t) \triangleq \frac{\mathrm{d}}{\mathrm{d}t}\left(q(\boldsymbol{X}_t,t)\right)_{|\boldsymbol{X}_t=\boldsymbol{x}} = 0, \tag{A3}$$

where the material derivative, $\frac{Dq}{\mathrm{d}t}$, (in Stratonovich notations) is simply:

$$\frac{Dq}{Dt} = \frac{\partial}{\partial t} + \dot{\boldsymbol{x}}_t \cdot \boldsymbol{\nabla}, \tag{A4}$$

with

$$\dot{\boldsymbol{x}}_t(\boldsymbol{x}) \triangleq \boldsymbol{w}^*(\boldsymbol{x},t) + \left(\boldsymbol{\sigma}\dot{\boldsymbol{B}}\right)(\boldsymbol{x},t). \tag{A5}$$

In Ito form, the transport equation (A3) makes explicit the turbulent diffusion and the centered anti-symmetric multiplicative
noise (Resseguier et al., 2017a)
We again highlight that this analysis holds for both SALT and LU approaches.





**Table A1.** Notation equivalences between LU and SALT approaches

|  | LU | SALT |
|---|---|---|
| Unresolved velocity | $(\boldsymbol{\sigma}\dot{\boldsymbol{B}})(\boldsymbol{x},t) =$ $\int_\Omega \mathrm{d}\boldsymbol{z}\,\breve{\boldsymbol{\sigma}}(\boldsymbol{x},\boldsymbol{z})\mathrm{d}\boldsymbol{B}_t(\boldsymbol{z})/\mathrm{d}t$ | $\sum_p \boldsymbol{\xi}_p(\boldsymbol{x})\mathrm{d}W_p(t)/\mathrm{d}t$ |
| Covariance | $\boldsymbol{\sigma}(\boldsymbol{x})\boldsymbol{\sigma}^T(\boldsymbol{y})/\mathrm{d}t =$ $\int_\Omega \mathrm{d}\boldsymbol{z}\,\breve{\boldsymbol{\sigma}}(\boldsymbol{x},\boldsymbol{z})\breve{\boldsymbol{\sigma}}^T(\boldsymbol{y},\boldsymbol{z})/\mathrm{d}t$ | $\sum_p \boldsymbol{\xi}_p(\boldsymbol{x})\boldsymbol{\xi}_p^T(\boldsymbol{y})/\mathrm{d}t$ |
| Variance tensor | $\boldsymbol{a}(\boldsymbol{x}) = \boldsymbol{\sigma}(\boldsymbol{x})\boldsymbol{\sigma}^T(\boldsymbol{x}) =$ $\int_\Omega \mathrm{d}\boldsymbol{z}\,\breve{\boldsymbol{\sigma}}(\boldsymbol{x},\boldsymbol{z})\breve{\boldsymbol{\sigma}}^T(\boldsymbol{x},\boldsymbol{z})$ | $\sum_p \boldsymbol{\xi}_p(\boldsymbol{x})\boldsymbol{\xi}_p^T(\boldsymbol{x})$ |
| Stratonovich drift | $\boldsymbol{w}^\star$ | $\boldsymbol{u}$ |
| Itō drift | $\boldsymbol{w} = \boldsymbol{w}^\star + \frac{1}{2}\left(\boldsymbol{\nabla}\cdot\boldsymbol{a}\right)^T$ | $\boldsymbol{u} + \frac{1}{2}\left(\boldsymbol{\nabla}\cdot\left(\sum_p \boldsymbol{\xi}_p\boldsymbol{\xi}_p^T\right)\right)^T$ |

## A4 Euler models

Nonetheless, the incompressible stochastic transports of velocity and vorticity differ between LU and SALT. First, the SALT approach considers the transport – through a specific Lagrangian choice and up to some forcings – of the Stratonovich large-scale linear momentum, $\rho\boldsymbol{w}^*$, whereas the LU Euler derivation assumes the transport of the Itō large-scale linear momentum,

$\rho\boldsymbol{w}$. Furthermore, due to its geometrical approach, the SALT Euler involves an additional term.

Specifically, the LU Euler equation with neither viscosity nor Coriolis force reads:

$$\frac{D}{Dt}\underbrace{\boldsymbol{w}}_{\substack{\text{Due to the}\\\text{transport}\\\text{of }\rho\boldsymbol{w}}} = -\frac{1}{\rho}\boldsymbol{\nabla}p, \tag{A6}$$

whereas the SALT version is:

$$\frac{D}{Dt}\underbrace{\boldsymbol{w}^*}_{\substack{\text{Due to the}\\\text{transport}\\\text{of }\rho\boldsymbol{w}^*}} + \underbrace{\boldsymbol{\nabla}\dot{\boldsymbol{x}}_t^T\boldsymbol{w}^*}_{\text{Additional term}} = -\frac{1}{\rho}\boldsymbol{\nabla}p. \tag{A7}$$

Unlike in the SALT model, the large-scale transported velocity of the LU Euler, $\boldsymbol{w}$, differs from the large-scale transporting velocity, $\boldsymbol{w}^*$. The latter implicitly appears in the stochastic transport operator $\frac{D}{Dt}$ of both the SALT and LU equations. Thus, in the LU equations, we can see the correction $\boldsymbol{w}^* - \boldsymbol{w} = -\frac{1}{2}\left(\boldsymbol{\nabla}\cdot\boldsymbol{a}\right)^T$ as a modification of the large-scale advection. Note that this modification cancels for a homogeneous small-scale velocity. Otherwise, whether the Ito drift :

$$\boldsymbol{w}(\boldsymbol{x},t) \triangleq \mathbb{E}\left(\boldsymbol{X}_{t+\mathrm{d}t} - \boldsymbol{X}_t | \boldsymbol{X}_t = \boldsymbol{x}\right)/\mathrm{d}t, \tag{A8}$$





or the Stratonovich drift:

$$\boldsymbol{w}^*(\boldsymbol{x},t) \overset{\triangle}{=} \mathbb{E}\left(\boldsymbol{X}_{t+\mathrm{d}t/2} - \boldsymbol{X}_{t-\mathrm{d}t/2}|\boldsymbol{X}_t = \boldsymbol{x}\right)/\mathrm{d}t, \tag{A9}$$

should be (randomly) transported is still an open question. A related question is how to interpret the "large-scale" velocity, derived from observations or from numerical simulations. Depending on the situation, this velocity may better correspond to the Itō drift or to the Stratonovich drift. Cotter et al. (2017) may help answer this question from a theoretical perspective.

On top of this large-scale advection difference between SALT and LU modeling, the SALT additional term, $\boldsymbol{\nabla}\dot{\boldsymbol{x}}_t^T \boldsymbol{w}^*$, has major consequences on the dynamics invariants, as explained in Sect. A6.

### A5   Other dynamical models

The differences between LU and SALT momentum equations – (A6) and (A7) respectively – resemble the distinctions between other classical and geophysical fluid dynamics models. Table A2 recalls some of these classical models in SALT and LU

stochastic formulations. The LU Euler equations can be found in Mémin (2014) without noise and in Resseguier (2017) including noise. The vorticity equation in each case is easily derived by taking the curl of the Euler equation. The LU QG equations under moderate influence of turbulence has been derived by Resseguier et al. (2017b) with Ito notation. In table A2, we have written the same equation in Stratonovich notation. The SALT Euler, vorticity and Quasi-Geostrophic (QG) equations are derived by Holm (2015).

Both SALT and LU assumes the stochastic transport of tracers (e.g. temperature, salinity). For QG models, it implies the transport of buoyancy, $b$, at the surface ($z = 0$). Therefore, up to the drift correction $\boldsymbol{w}^* - \boldsymbol{w}$ in the streamfunction definition, SALT and PV Potential Vorticity (PV) coincide. In the SQG framework, the PV is assumed to be zero in the ocean interior ($z < 0$). This leads to the same relationship between buoyancy and streamfunction, $\psi$, in LU and SALT dynamics. Therefore, in the Sect. 3 of this paper, we will choose this model to compare two parameterization (i.e. choice of $\boldsymbol{\sigma}$) of SALT-LU method

in a common ground.

### A6   Invariants

To interpret these results which apply more generally, mutatis mutandis the KE is conserved by the LU scheme and its variants (Resseguier et al., 2017a) while the enstrophy, its generalizations (e.g., PV), the circulation and the helicity are conserved by the SALT scheme and its variants (Holm, 2015) . In the specific case of homogeneous small-scale velocity, the LU dynamics

also conserve circulation and helicity in average (see appendix A7 for circulation mean conservation; the proof of helicity mean conservation is similar). However, in the homogeneous case, this stochastic dynamics increases enstrophy mean and its generalizations (Resseguier et al., 2017b) while SALT models increase KE mean (Resseguier, 2017).

These distinctions between conservation of vorticity in the SALT approach and conservation of energy in the LU approach would be of critical importance when choosing a model for 2D or QG turbulence, as Kraichnan (1967); Charney (1971) show

that the conservation of energy and enstrophy lead to turbulence typified by an inverse energy cascade at large scales and a direct (potential) enstrophy cascade at small scales (Fox-Kemper and Menemenlis, 2008). In SQG, Blumen (1982) shows that





| Equations | LU | SALT |
|---|---|---|
| 3D Euler eq. in a rotating frame (with 3D $\boldsymbol{\nabla}$) | $\rho\left(\frac{D}{Dt}+\boldsymbol{f}\times\right)\boldsymbol{w}$ $+\boldsymbol{f}\times\boldsymbol{\sigma}\dot{\boldsymbol{B}}=(\rho\boldsymbol{g}-\boldsymbol{\nabla}p),$ $D_t\rho=0.$ | $\rho\left(\frac{D}{Dt}+\boldsymbol{\nabla}\dot{\boldsymbol{x}}_t^T+\boldsymbol{f}\times\right)\boldsymbol{w}^\star$ $+\boldsymbol{f}\times\boldsymbol{\sigma}\dot{\boldsymbol{B}}=(\rho\boldsymbol{g}-\boldsymbol{\nabla}p),$ $D_t\rho=0.$ |
| Incompressible 3D vorticity eq. (with 3D $\boldsymbol{\nabla}$) | $\boldsymbol{\omega}=\boldsymbol{\nabla}\times\boldsymbol{w},$ $\frac{D\boldsymbol{\omega}}{Dt}=(\boldsymbol{\omega}\cdot\boldsymbol{\nabla})\dot{\boldsymbol{x}}_t$ $-\sum_{q=1}^d\boldsymbol{\nabla}(\dot{\boldsymbol{x}}_t)_q\times\boldsymbol{\nabla}w_q,$ $\boldsymbol{\nabla}\cdot\boldsymbol{w}^\star=0,\ \boldsymbol{\nabla}\cdot\boldsymbol{\sigma}=0.$ | $\boldsymbol{\omega}^\star=\boldsymbol{\nabla}\times\boldsymbol{w}^\star,$ $\frac{D\boldsymbol{\omega}^\star}{Dt}=(\boldsymbol{\omega}^\star\cdot\boldsymbol{\nabla})\dot{\boldsymbol{x}}_t,$ $\boldsymbol{\nabla}\cdot\boldsymbol{w}^\star=0,\ \boldsymbol{\nabla}\cdot\boldsymbol{\sigma}=0.$ |
| Incompressible 2D vorticity eq. (with 2D $\boldsymbol{\nabla}$) | $\zeta=\boldsymbol{\nabla}^\perp\cdot\boldsymbol{w},$ $\frac{D\zeta}{Dt}=-\text{tr}\left(\boldsymbol{\nabla}^\perp\dot{\boldsymbol{x}}_t^T\boldsymbol{\nabla}\boldsymbol{w}^T\right),$ $\boldsymbol{\nabla}\cdot\boldsymbol{w}^\star=0,\ \boldsymbol{\nabla}\cdot\boldsymbol{\sigma}=0.$ | $\zeta^\star=\boldsymbol{\nabla}^\perp\cdot\boldsymbol{w}^\star,$ $\frac{D\zeta^\star}{Dt}=0,$ $\boldsymbol{\nabla}\cdot\boldsymbol{w}^\star=0,\ \boldsymbol{\nabla}\cdot\boldsymbol{\sigma}=0.$ |
| Quasi-geostrophic equations (with 2D $\boldsymbol{\nabla}$, $\Delta$) | $\boldsymbol{w}=\boldsymbol{\nabla}^\perp\psi,\ \boldsymbol{\nabla}\cdot\boldsymbol{w}^\star=0,\ \boldsymbol{\nabla}\cdot\boldsymbol{\sigma}=0,$ $PV=\Delta\psi+f+(f_0/N)^2\partial_z^2\psi,$ $f_0\partial_z\psi=b,$ $\frac{DPV}{Dt}=(\boldsymbol{w}^*-\boldsymbol{w})\cdot\boldsymbol{\nabla}f\text{d}t$ $-\text{tr}\left(\boldsymbol{\nabla}^\perp\dot{\boldsymbol{x}}_t^T\boldsymbol{\nabla}\boldsymbol{w}^T\right),\text{ for }z<0$ $\frac{Db}{Dt}=0\text{ for }z=0.$ | $\boldsymbol{w}^\star=\boldsymbol{\nabla}^\perp\psi,\ \boldsymbol{\nabla}\cdot\boldsymbol{\sigma}=0,$ $PV=\Delta\psi+f+(f_0/N)^2\partial_z^2\psi,$ $f_0\partial_z\psi=b,$ $\frac{DPV}{Dt}=0,\text{ for }z<0$ $\frac{Db}{Dt}=0\text{ for }z=0.$ |
| Surface quasi-geostrophic eq. (with 2D $\boldsymbol{\nabla}$, $\Delta$) | $\boldsymbol{w}=\boldsymbol{\nabla}^\perp\psi,\ \boldsymbol{\nabla}\cdot\boldsymbol{w}^\star=0,\ \boldsymbol{\nabla}\cdot\boldsymbol{\sigma}=0,$ $(-\Delta)^{1/2}\psi=\frac{1}{N}b,$ $\frac{Db}{Dt}=0.$ | $\boldsymbol{w}^\star=\boldsymbol{\nabla}^\perp\psi,\ \boldsymbol{\nabla}\cdot\boldsymbol{\sigma}=0,$ $(-\Delta)^{1/2}\psi=\frac{1}{N}b,$ $\frac{Db}{Dt}=0.$ |

**Table A2.** Some LU and SALT dynamics models with Stratonovich notations.





a dual cascade results from conservation of depth-integrated energy and available potential energy on level boundaries. The former is not singled out for special treatment in the SALT or LU SQG formulation, but the latter is exactly conserved in both formulations.

### A7 Kelvin theorem under location uncertainty

The Kelvin theorem describes the variation of the circulation, which is defined as follows for LU dynamics:

$$\Gamma = \oint_{C(t)} \boldsymbol{w}(\boldsymbol{x},t) \cdot d\boldsymbol{l} = \oint_{C(0)} \boldsymbol{w}(\boldsymbol{X}_t(\boldsymbol{l}_0),t)^T \boldsymbol{J}_M \, d\boldsymbol{l}_0, \tag{A10}$$

where $\boldsymbol{l} = \boldsymbol{X}_t(\boldsymbol{l}_0)$ is the Lagrangian path labeled by the initial position $\boldsymbol{l}_0$, $C(t) = \boldsymbol{X}_t(C(0))$ is a material loop at time $t$ and $\boldsymbol{J}_M = \boldsymbol{J}_M(\boldsymbol{l}_0,t) = [\boldsymbol{\nabla}\boldsymbol{X}_t^T]^T(\boldsymbol{l}_0)$ is the Jacobian matrix of the flow. The time differentiation of the circulation involves the

time variation of the Jacobian matrix $(\mathrm{d}\boldsymbol{J}_M = [\boldsymbol{\nabla}\dot{\boldsymbol{x}}_t^T]^T \boldsymbol{J}_M)$:

$$\frac{\mathrm{d}\Gamma}{\mathrm{d}t} = \oint_{C(0)} \left( \frac{D\boldsymbol{w}}{Dt}^T \boldsymbol{J}_M + \boldsymbol{w}^T [\boldsymbol{\nabla}\dot{\boldsymbol{x}}_t^T]^T \boldsymbol{J}_M \right) d\boldsymbol{l}_0 = \oint_{C(t)} \left( -\tfrac{1}{\rho}\boldsymbol{\nabla}p + [\boldsymbol{\nabla}\dot{\boldsymbol{x}}_t^T]\boldsymbol{w} \right) \cdot d\boldsymbol{l} = \oint_{C(t)} (\boldsymbol{\nabla}\dot{\boldsymbol{x}}_t^T \boldsymbol{w}) \cdot d\boldsymbol{l}. \tag{A11}$$

This equation is the equivalent of the Reynolds transport theorem for vorticity Fox-Kemper and Pedlosky (2004), but in the stochastic framework. This noise term is *a priori* not centered since it is a Stratonovich noise. However, in the homogeneous case, its ensemble mean cancels. Indeed, using Ito notations, we simply need to compute a quadratic cross-variation. This

calculus is possible in Lagrangian coordinates (i.e. when functions are composed by $\boldsymbol{x} \mapsto \boldsymbol{X}_t$) by noticing that $\frac{D\boldsymbol{w}}{Dt}$ is a term in $\mathrm{d}t$, $\frac{D\boldsymbol{\sigma}}{Dt} = (\dot{\boldsymbol{x}}_t \cdot \boldsymbol{\nabla})\boldsymbol{\sigma}$ and $\frac{D\boldsymbol{J}_M}{Dt} = \frac{\mathrm{d}\boldsymbol{J}_M}{\mathrm{d}t} = [\boldsymbol{\nabla}\dot{\boldsymbol{x}}_t^T]^T \boldsymbol{J}_M$ :

$$\frac{\mathrm{d}}{\mathrm{d}t}\mathbb{E}\{\Gamma\} = \frac{1}{\mathrm{d}t}\mathbb{E}\oint_{C(0)} \left( \boldsymbol{w}^T [\boldsymbol{\nabla}(\boldsymbol{\sigma} \circ \mathrm{d}\boldsymbol{B}_t)^T]^T \boldsymbol{J}_M \right) d\boldsymbol{l}_0, \tag{A12}$$

$$= \tfrac{1}{2}\mathbb{E}\oint_{C(0)} \sum_k \frac{D}{Dt} \left\langle \boldsymbol{w}^T [\boldsymbol{\nabla}\boldsymbol{\sigma}_{\bullet k}^T]^T \boldsymbol{J}_M d\boldsymbol{l}_0, (\boldsymbol{B}_t)_k \right\rangle, \tag{A13}$$

$$= \tfrac{1}{2}\mathbb{E}\oint_{C(0)} \sum_k \boldsymbol{w}^T \left( (\boldsymbol{\sigma}_{\bullet k} \cdot \boldsymbol{\nabla}) + [\boldsymbol{\nabla}\boldsymbol{\sigma}_{\bullet k}^T]^T \right) [\boldsymbol{\nabla}\boldsymbol{\sigma}_{\bullet k}^T]^T \boldsymbol{J}_M d\boldsymbol{l}_0, \tag{A14}$$

$$= \tfrac{1}{2}\mathbb{E}\oint_{C(t)} \sum_{pqk} w_q \boldsymbol{\nabla} \cdot \partial_p \underbrace{(\sigma_{qk}\boldsymbol{\sigma}_{\bullet k})}_{\text{cst.}} d\boldsymbol{l}_p, \tag{A15}$$

$$= 0. \tag{A16}$$

Therefore, the mean LU circulation is conserved in the homogeneous case.

### Appendix B: Effective resolution and inertial range

Let us assume the simulated evolution law is $D_t b = -\nu(-\Delta)^p b \, \mathrm{d}t$. The deterministic subgrid model $-\nu(-\Delta)^p b$ acts, in a

finite time $t$, as a low-pass filter. In Fourier space, this filter is:

$$F(\|\boldsymbol{k}\|) = \exp\left(-\nu t\|\boldsymbol{k}\|^{2p}\right). \tag{B1}$$





If the hyperviscosity $\nu$ is well chosen, we may expect that at the Shannon resolution $\pi/\Delta x = \kappa_M$, only $10\%$ of the energy is left by the filter, i.e.

$$F(\kappa_M) = 1/10. \tag{B2}$$

A ratio smaller than $10\%$ may lead to an over-damped simulation. Moreover, the precise value of this ratio does not influence much our final estimate.

We may define the effective resolution as the scale $\kappa = \kappa_m$ where the deterministic subgrid model influence is negligible. There, we may expect the filter to be equal to $95\%$, i.e.:

$$F(\kappa_m) = 95/100. \tag{B3}$$

The ratio $\kappa_m/\kappa_M$ can then be derived from formulas (B1), (B2) and (B3).



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
