# Peer review of "Data-driven versus self-similar parameterizations for Stochastic Advection by Lie Transport and Location Uncertainty"

_Nonlinear Processes in Geophysics, 2019_

## Referee Comment (RC1) · Anonymous Referee #1 · 26 Nov 2019

In this study, the authors focused on the common challenge of the stochastic subgrid parameterization schemes: the unresolved velocity construction. Two kinds of parameterizations, data-driven and self-similar parameterizations, were applied to LU and SALT frameworks. The results show that these two parameterizations can lead to high quality ensemble forecasts. In my opinion, the main innovation of this study is the proposal of the self-similar parameterization, which improves the work of Resseguier et al. (2017b). Although this manuscript may be suitable for publication in NPG, there are still some issues to be addressed.

1. Recently, many parameterizations are available. In this study, the authors proposed

a new self-similar parameterization. I know that its advantage is tuning-free. However, more interestingly, when this parameterization is used to the numerical models, whether the improvements of the simulations or forecast are significantly enhanced, comparing to other parameterizations, especially for the one of Resseguier et al. (2017b).

2. Figure 7 shows that the patterns obtained by the data-driven and self-similar parameterizations are similar to that in the Low-resolution deterministic SQG model at day 110. This means that, for the short-term simulations, the stochastic subgrid parameterizations have very weak improvements on the low-resolution simulations?

3. The authors tested the two parameterizations in the SQG model. This model is very simple. Please discuss how to apply these parameterizations to the complicated atmosphere and ocean models.

4. In this study, the term "SALT-LU" appears frequently. In my opinion, this term may mislead readers. They can think that the authors aimed at combining the SALT and LU parameterizations.

5. Lines 47 and 51. in (Gay-Balmaz and Holm, 2018)—-> in Gay-Balmaz and Holm (2018); in (Cotter et al., 2018b, a) —-> in Cotter et al. (2018b, a)

6. Line 126. Two vertical lines were not plotted in the left panel of figure 1.

7. Line 401. Similar results UQ results—-> Similar UQ results?

---

## Referee Comment (RC2) · Anonymous Referee #2 · 13 Dec 2019

General: This paper introduced the stochastic subgrid parameterizations which express the unresolved velocity from the large-scale velocity. Self-similar schemes with SALT and LU frameworks are proposed with details and compared with the data-driven models in the Surface Quasi-Geostrophic (SQG) model. The authors focus on their common challenge: the parameterization choice. The results show that both parameterizations lead to high quality ensemble forecasts. This paper is well organized, and also contains some interesting components. I think it is suitable for publication in NPG, however, there are some issues to be addressed.

1.Page 6, the authors compared figures 2 and 3 in line 171, indicated some features

of the homogeneous parameterization in line 174. Actually, only the first row of figure 3 was referred here. In Page 8 line 205, they state 'Figure 3 confirms that this modulation enables a more accurate spatial distribution of the stochastic foldings', it is more convincing if the reference distribution is in the same figure. I suggest to merge figure 2 and 3.

2.Page 19, line 370 and line 378, it is not recommended to say SALT-LU SQG dynamics. Although the slight difference between the SQG SALT and the SQG LU models are not considered in this section (indicated in line 318), they are not combined. So, please use an other abbreviate.

3.Page 16, the caption of figure 4 is not correct.

4.A legend is required for the right plot of figure 6. Also the colors for different number of EOFs are difficult to distinguish in that plot.

5.Figure 7, although the SLAT/LU dynamics is not worse than a low-resolution deterministic simulation, it did not show many advantages as figures 2&3 shown. The scenarios of figure 7 and figure2&3 have different resolutions (128ˆ2 and 64ˆ2) and different integration days(day 15 and day 110). Give reasons why both data-driven and self-similar parameterizations have very weak improvements in figure 7.

6.Page 22, line 393, why the error estimation is 1.96 times the point-wise std?

7.Mark the grid points chosen for figures 10 and 11 in figure 9.

8.The authors have shown the predictability time for one realization is about 2 weeks. They also showed that the ensemble forecast can capture well the reference dynamics of the center of the ensemble distribution for longer period in figures 10 and 11. It is hard to tell the predictability time for ensemble forecast from the first column of figure 9. The authors should plot the ensemble mean of each SQG dynamics analogs to figures 7 and 8, and make a statement about the predictability time of ensemble forecast.

---

## Author Comment (AC1) · 9 Jan 2020

**Answer to reviewer #1**

First of all, we would like to express our warm thanks to the reviewers and the editor for the evaluation work they did and for all the comments and suggestions the reviewers have provided on our study. Hereafter, we will answer in details the questions of the referee #1. Accordingly, we will propose some changes that we could do in a revised version, if the editor enables a revised submission.

*"In this study, the authors focused on the common challenge of the stochastic subgrid parameterization schemes: the unresolved velocity construction. Two kinds of parameterizations, data-driven and self-similar parameterizations, were applied to LU and SALT frameworks. The results show that these two parameterizations can lead to high quality ensemble forecasts. In my opinion, the main innovation of this study is the proposal of the self-similar parameterization, which improves the work of Resseguier et al. (2017b)."*

Thank you for this summary of our work.

*"Although this manuscript may be suitable for publication in NPG, there are still some issues to be addressed."*

1. *"Recently, many parameterizations are available. In this study, the authors proposed a new self-similar parameterization. I know that its advantage is tuning-free. However, more interestingly, when this parameterization is used to the numerical models, whether the improvements of the simulations or forecast are significantly enhanced, comparing to other parameterizations, especially for the one of Resseguier et al. (2017b)."*

   For stationary, fully-developed turbulence and after including a tuning stage to optimize the match, the self-similar parameterization and the method of Resseguier et al. (2017b) give approximately the same results. Therefore, in order to appreciate improvements of the simulations or of the forecasts, we must either work with non-stationary flows or with erroneous tuning. This latter scenario is of course the meaningful one for the application of such models to the real world, where turbulence is non-stationary, heterogeneous, and so region-by-region and season-by-season tuning is impossible to do with quality checks in place.

   We believe that non-stationarity may yield a fair, yet idealized, comparison. So, in a revised manuscript, we propose to compare the two parameterizations in a

non-stationary case. The figure 1 below shows simulated buoyancy fields initialized with the "case 2" of Constantin et al. (1994) and corresponding errors. After two days of advection, there is no turbulence yet, and a $128\text{x}128$ resolution is sufficient to correctly resolve every scale. Therefore, no stochastic subgrid parameterization is needed. The self-similar method automatically adapts to the situation, whereas the method of Resseguier et al. (2017b) introduces spurious buoyancy isolines roughness by randomly folding these isolines. Accordingly, the method of Resseguier et al. (2017b) introduces more errors.

2. **"Figure 7 shows that the patterns obtained by the data-driven and self-similar parameterizations are similar to that in the Low-resolution deterministic SQG model at day 110. This means that, for the short-term simulations, the stochastic subgrid parameterizations have very weak improvements on the low-resolution simulations?"**

Yes, this is right. For the short-term simulations, our stochastic subgrid parameterizations have often weak improvements on the low-resolution simulations, even though, sometimes, the stochastic subgrid parameterization can improve the simulation. Indeed, Resseguier et al. (2017b) show that the LU dynamics at a resolution $128\text{x}128$ can trigger filament instabilities by random destabilization, and hence obtain a more realistic proportion of eddies and filaments. This is confirmed by the figures 2 and 3 of our submitted draft, also at a resolution $128\text{x}128$. In figure 7, the resolution is coarser ($64\text{x}64$). Therefore, the stabilizing deterministic subgrid tensor (hyper viscosity) is stronger. This may explain an inhibition of filament instabilities here, and hence less difference between deterministic and stochastic coarse simulations.

We could add this discussion to the subsection "3.3 One realization" in a revised version of our draft.

Nevertheless, our main goal is not improving a single simulation. Our main goal is improving the uncertainty quantification without deteriorating single simulations.

3. **"The authors tested the two parameterizations in the SQG model. This model is very simple. Please discuss how to apply these parameterizations to the complicated atmosphere and ocean models."**

It is true that SQG is a simple model, in particular because the vertical variations are described by analytic formulas and hence do not need to be simulated. The reason for the selection of this model was that it occurs identically in the SALT and LU formulations, not that this is an especially realistic model. For more complicated atmosphere and ocean models, such as hydrostatic Boussinesq models, simplicity is not to be found. The vertical dimension imposes anisotropy and heterogeneity of the small-scale velocity, at least breaking symmetric between the vertical and horizontal directions. Thus anisotropy and heterogeneity need to be parameterized–in the SQG system the anisotropy and vertical heterogeneity is prescribed. Moreover, complex boundary conditions and heterogeneity often prevent the direct use of Fourier transform. We develop these two points in the following.

(a) Third dimension, anisotropy and vertical heterogeneity :
In order to impose the divergence-free condition ($\boldsymbol{\nabla} \cdot \dot{B} = 0$) in 3D, one can build the small-scale velocity $\sigma \dot{B}$ from a 3D curl $\boldsymbol{\nabla}_{3D} \times$ and a 3D streamfunction $\phi_\sigma \dot{B}$:

$$\boldsymbol{\sigma}\dot{\boldsymbol{B}} = \boldsymbol{\nabla}_{3D} \times \left(\phi_\sigma \dot{\boldsymbol{B}}\right). \qquad (1)$$

This streamfunction can be filter along $z$ in order to impose a given finite vertical correlation length $H_\sigma$ related to the vertical resolution (e.g. $H_\sigma = 2\Delta z$) :

$$\phi_\sigma \dot{\boldsymbol{B}}(x, y, z) = \breve{\phi}_\sigma^z(z) *_z \left(\left(\phi_\sigma^{2D} \dot{\boldsymbol{B}}\right)(x, y, z)\right), \qquad (2)$$

with $*_z$ a one-dimensional convolution along $z$ and $\check{\phi}_\sigma^z$ a one-dimensional vertical Gaussian filter:

$$\check{\phi}_\sigma^z(z) = (2\pi)^{1/4} H_\sigma^{1/2} \exp\left(-\frac{z^2}{H_\sigma^2}\right).$$ (3)

For simplicity, the 3 components of the horizontal streamfunction $\phi_\sigma^{2D}\dot{B}$ can be assumed to be statistically independent. Each of these $z$-dependent component can be defined – at each depth $z$ independently – similarly to the horizontal streamfunction of our draft:

$$\left(\phi_\sigma^{2D}\dot{B}\right)_k(x,y,z) = \check{\phi}_\sigma^{2D}(x,y,z) *_{(x,y)} \dot{B}_k(x,y,z),$$ (4)

for $1 \leqslant k \leqslant 3$, where $*_{(x,y)}$ is a horizontal two-dimensional convolution, $\dot{B}$ is 3D spatio-temporal white noise, $\check{\phi}_\sigma^{2D}$ is a scalar horizontal spatial filter. If Fourier transforms are possible, that filter can be defined as in our draft at each depth $z$ from the horizontal large-scale resolved velocity statistics. Otherwise, that filter can be a 2D Matérn covariance (Williams and Rasmussen, 2006; Lim and Teo, 2009; Lilly et al., 2017) of a correlation length $2\pi/\kappa_m$ related to the horizontal resolution (e.g. $\pi/\kappa_m = 2\sqrt{\Delta x \Delta y}$) and a regularity parameter $(r(z)+1)/2$ learned from the horizontal large-scale resolved velocity. Indeed, with such a parameterization, for large horizontal wave number $\kappa^{2D} = \sqrt{k_x^2 + k_y^2}$, the horizontal ADSD is a smooth-along-$z$ version of :

$$cst. \left(\frac{\kappa^{2D}}{\kappa_m}\right)^{-r(z)}.$$ (5)

Therefore, self-similar arguments can be still be used to identify the regularity $(r(z)+1)/2$ at large and small scales.

(b) Boundary conditions and heterogeneity :

Non-periodic boundary conditions suggest small-scale velocity heterogeneity, at least in the variance. For instance, Dirichlet boundary conditions for the velocity suggest that the small-scale velocity variance should be zero on the boundary. Since this variance is non-zero inside the domain, there are variance heterogeneity.

A first solution is to ignore the small-scale velocity boundary conditions, as in (Chapron et al., 2018). Indeed, since that velocity appears only in a multiplicative way, the small-scale velocity homogeneity will be modulated by the transported resolved fields heterogeneity. Therefore, the erroneous small-scale velocity homogeneity is expected to have only weak impact.

If one wants to impose small-scale boundary conditions anyway, the homogeneous small-scale velocity $v' = \sigma \dot{B}$ – defined above – can be "conditioned on" the values of the boundary conditions. As an example, we can consider a one-dimensional process $v'(x)$, with an (unconditioned) covariance $\gamma_{v'}$ and a correlation length (small compared to the domain size $L$). A velocity $v'_{BC}$ which respect the boundary conditions $v'_{BC}(0) = v'_0$ and $v'_{BC}(L) = v'_L$ is the conditioning of velocity $v'$ on the boundary conditions. That conditioned velocity can be simulated as:

$$v'_{BC}(x) = v'(x) - \frac{\gamma_{v'}(x-0)}{\gamma_{v'}(0)}\left(v'(0) - v'_0\right) - \frac{\gamma_{v'}(x-L)}{\gamma_{v'}(0)}\left(v'(L) - v'_L\right). \quad (6)$$

Nevertheless, we believe that the above discussion is overly complex and not mature enough to add it to our draft. Applying these parameterizations to the complicated atmosphere and ocean models would require future works.

4. **"In this study, the term "SALT-LU" appears frequently. In my opinion, this term may mislead readers. They can think that the authors aimed at combining the SALT and LU parameterizations."**

In a revised version, we propose to use instead either only "LU" or "SALT and LU", depending on the readability of the sentence.

5. **"Lines 47 and 51. in (Gay-Balmaz and Holm, 2018) → in Gay-Balmaz and Holm (2018); in (Cotter et al., 2018b, a) → in Cotter et al. (2018b, a)"**

   We will correct this.

6. **"Line 126. Two vertical lines were not plotted in the left panel of figure 1."**

   We will correct this.

7. **"Line 401. Similar results UQ results → Similar UQ results?"**

   This is also a typo and we will correct it.

We thank again the reviewer for all these useful comments and questions.

Figure 1: Buoyancy field (m.s$^{-2}$) (top) and corresponding normalized error (dimensionless) (bottom) after one day of advection for the 1024x1024 reference deterministic simulation (left), the 128x128 LU simulation with self-similar parameterization (middle), the 128x128 LU simulation with the method of Resseguier et al. (2017) (right).

**Fig. 1.**

---

## Author Comment (AC2) · 9 Jan 2020

**Answer to reviewer #2**

First of all, we would like to express our warm thanks to the reviewers and the editor for the evaluation work they did and for all the comments and suggestions the reviewers have provided on our study. Hereafter, we will answer in details the questions of the referee #2. Accordingly, we will propose some changes that we could do in a revised version, if the editor enables a revised submission.

[Figure]

"**General: This paper introduced the stochastic subgrid parameterizations which express the unresolved velocity from the large-scale velocity. Self-similar schemes with SALT and LU frameworks are proposed with details and compared with the data-driven models in the Surface Quasi-Geostrophic (SQG) model. The authors focus on their common challenge: the parameterization choice. The results show that both parameterizations lead to high quality ensemble forecasts.**"

Thank you for this summary of our work.

"**This paper is well organized, and also contains some interesting components.**"

Thank you for the interest.

"**I think it is suitable for publication in NPG, however, there are some issues to be addressed.**"

1. "**Page 6, the authors compared figures 2 and 3 in line 171, indicated some features of the homogeneous parameterization in line 174. Actually, only the first row of figure 3 was referred here. In Page 8 line 205, they state 'Figure 3 confirms that this modulation enables a more accurate spatial distribution of the stochastic foldings', it is more convincing if the reference distribution is in the same figure. I suggest to merge figure 2 and 3.**"

   You are right, it would be clearer if figures 2 and 3 are merged. We have separated them because the figure was originally too large for the page. Nevertheless, we can try to merge them again in a revised version.

2. "**Page 19, line 370 and line 378, it is not recommended to say SALT-LU SQG dynamics. Although the slight difference between the SQG SALT and the**"

**SQG LU models are not considered in this section (indicated in line 318), they are not combined. So, please use an other abbreviate."**

In a revised version, we propose to use instead either only "LU" or "SALT and LU", depending on the readability of the sentence.

3. **"Page 16, the caption of figure 4 is not correct."**

Yes this is right. We have written :

"Buoyancy (m.s$^{-2}$) at $t = 0, 10, 30, 50$ and $70$ days of advection (top) and its spectrum (m$^2$.s$^{-4}$/(rad.m$^{-1}$)) at $t = 50$ days of advection (bottom) for the deterministic SQG model at resolution $512^2$."

Instead of :

"Buoyancy (m.s$^{-2}$) (left), KE spectrum (m$^2$.s$^{-2}$/(rad.m$^{-1}$)) (middle) and ADSDs (m$^2$.s$^{-1}$/(rad.m$^{-1}$)) (right) at $t = 0, 30, 50$ and $70$ days of advection, for the deterministic SQG model at resolution $512^2$."

We will correct that.

There is also a typo in the caption of figure 6 – "$t = 17$ days" instead of "$t = 100$ days" – that will be corrected.

4. **"A legend is required for the right plot of figure 6. Also the colors for different number of EOFs are difficult to distinguish in that plot."**

Indeed, the plot is not easy to read. We will add a legend and try colors which are more different.

5. **"Figure 7, although the SALT-LU dynamics is not worse than a low-resolution deterministic simulation, it did not show many advantages as figures 2&3 shown. The scenarios of figure 7 and figure 2&3 have different resolutions ($128^2$ and $64^2$) and different integration days(day 15 and day**

**110). Give reasons why both data-driven and self-similar parameterizations have very weak improvements in figure 7."**

For the short-term simulations, our stochastic subgrid parameterizations have often weak improvements on the low-resolution simulations, even though, sometimes, the stochastic subgrid parameterization can improve the simulation. Indeed, Resseguier et al. (2017b) show that the LU dynamics at a resolution $128x128$ can trigger filament instabilities by random destabilization, and hence obtain a more realistic proportion of eddies and filaments. This is confirmed by the figures 2 and 3 of our submitted draft, also at a resolution $128x128$. In figure 7, the resolution is coarser $(64x64)$. Therefore, the stabilizing deterministic subgrid tensor (hyper viscosity) is stronger. This may explain an inhibition of filament instabilities here, and hence less difference between deterministic and stochastic coarse simulations.

We could add this discussion to the subsection "3.3 One realization" in a revised version of our draft.

Nevertheless, our main goal is not improving a single simulation. Our main goal is improving the uncertainty quantification without deteriorating single simulations.

6. **"Page 22, line 393, why the error estimation is 1.96 times the point-wise std?"**

Mean $+/-1.96$ times the point-wise std bound the $95\%$-confidence interval for Gaussian variables. Here, the buoyancy is not Gaussian. However, we believe that $+/-1.96$ times the point-wise std remains a simple and convenient approximate metric to define an acceptable bias.

We should explain more this choice in a revised version.

7. **"Mark the grid points chosen for figures 10 and 11 in figure 9."**

We will do this.

8. **"The authors have shown the predictability time for one realization is about 2 weeks. They also showed that the ensemble forecast can capture well the reference dynamics of the center of the ensemble distribution for longer period in figures 10 and 11. It is hard to tell the predictability time for ensemble forecast from the first column of figure 9. The authors should plot the ensemble mean of each SQG dynamics analogs to figures 7 and 8, and make a statement about the predictability time of ensemble forecast."**

We will add these plots in a revised version.

We thank again the reviewer for all these useful comments and questions.